# Enhancing plasticity in central networks improves motor and sensory recovery after nerve damage

Eric C. Meyers [1]*, Nimit Kasliwal [1], Bleyda R. Solorzano[1], Elaine Lai[2], Geetanjali Bendale[3], Abigail Berry[1], Patrick D. Ganzer[1], Mario Romero-Ortega [1,2,3], Robert L. Rennaker II[1,2,3], Michael P. Kilgard[1,2,3] & Seth A. Hays[1,2,3]

Nerve damage can cause chronic, debilitating problems including loss of motor control and paresthesia, and generates maladaptive neuroplasticity as central networks attempt to compensate for the loss of peripheral connectivity. However, it remains unclear if this is a critical feature responsible for the expression of symptoms. Here, we use brief bursts of closed-loop vagus nerve stimulation (CL-VNS) delivered during rehabilitation to reverse the aberrant central plasticity resulting from forelimb nerve transection. CL-VNS therapy drives extensive synaptic reorganization in central networks paralleled by improved sensorimotor recovery without any observable changes in the nerve or muscle. Depleting cortical acetylcholine blocks the plasticity-enhancing effects of CL-VNS and consequently eliminates recovery, indicating a critical role for brain circuits in recovery. These findings demonstrate that manipulations to enhance central plasticity can improve sensorimotor recovery and define CL-VNS as a readily translatable therapy to restore function after nerve damage.

---

[1] Texas Biomedical Device Center, The University of Texas at Dallas, 800 West Campbell Road, Richardson, TX 75080-3021, USA. [2] School of Behavioral and Brain Sciences, The University of Texas at Dallas, 800 West Campbell Road, Richardson, TX 75080-3021, USA. [3] Department of Bioengineering, Erik Jonsson School of Engineering and Computer Science, The University of Texas at Dallas, 800 West Campbell Road, Richardson, TX 75080-3021, USA. *email: eric.meyers@utdallas.edu

Nerve damage is a debilitating neurological disorder that affects over one hundred million people worldwide[1]. Most traumatic nerve injuries occur in the upper extremities, resulting in profound motor and sensory loss and chronic dysfunction of the arm and hand, which can severely reduce quality of life[2]. Despite advances in surgical repair and scaffolding techniques that promote regeneration, long-term prognosis for full recovery of normal sensation and muscle control is poor[3,4]. There is a clear and present need to develop interventional strategies that target alternative mechanisms beyond nerve regeneration to improve recovery of motor and sensory function after nerve injury[5].

Immediately after traumatic nerve damage, physical disconnection results in loss of motor and sensory function. Even after successful nerve regeneration, peripheral damage precipitates lasting changes throughout the central nervous system, as injured axons in the damaged nerve sprout, regrow, and establish new connections with both appropriate and aberrant targets[5,6]. This includes extensive reorganization of synaptic connectivity in regions of the brain and spinal cord that control motor and sensory function. The profound and long-lasting reorganization of these central networks attempts to compensate for altered peripheral connectivity caused by reinnervation errors[7–11]. Pioneering studies conducted over three decades ago demonstrated that synaptic connections from spared circuits strengthen and dominate central network activity in the absence of competition from the denervated circuits[8–10,12–14]. Therefore, despite the reconnection of peripheral axons to end targets (e.g., reinnervated muscle fibers and sensory receptors), chronic dysfunction of motor control and sensation often persists. The inability for weakened reinnervated networks to overcome this maladaptive central plasticity that occurs in response to nerve damage may contribute to lasting dysfunction[15,16].

Indirect evidence supports the role of maladaptive central plasticity in chronic dysfunction following nerve injury. Compared to adults, children often display greater recovery despite a similar degree of reinnervation inaccuracies after nerve injury, an effect partially attributed to a greater capacity for central plasticity in children[15,17]. Moreover, imaging studies in humans reveal long-term changes in brain structure and network function that associate with functional impairment[5,16,18,19]. Thus, if maladaptive central plasticity after nerve damage contributes to chronic dysfunction, then techniques that reestablish normal central network signaling should improve function, even in the absence of changes to the damaged nerve itself.

In this study, we directly test the hypothesis that reversing the maladaptive plasticity that occurs in response to nerve damage will support motor and sensory recovery. We leverage a closed-loop neuromodulation strategy using vagus nerve stimulation (CL-VNS) to provide precisely-timed release of neuromodulators, including acetylcholine, during rehabilitation[20–22]. CL-VNS has previously been used to enhance central plasticity and improve function in the context of stroke and spinal cord injury[23–26]. Rats underwent complete transection and tubular repair of the median and ulnar nerves in the forelimb. Six weeks after nerve repair, animals undergo rehabilitative training with short bursts of vagus nerve stimulation delivered coincident with successful pull attempts. Two groups that either decouple VNS from rehabilitation or deplete acetylcholine in the brain were included to control for VNS effects independent of central plasticity, including direct effects on regeneration and target reinnervation. We observe that CL-VNS reverses the maladaptive expansion of cortical circuits resulting from nerve damage and restores the descending drive of injured networks in the absence of observable effects on peripheral nerve or muscle health. These changes are paralleled by the enhancement of motor and sensory recovery.

These data demonstrate a causal role for central plasticity in dysfunction after nerve injury and introduce CL-VNS paired with rehabilitation as a readily translatable therapy.

## Results

**CL-VNS reverses maladaptive plasticity**. To assess the consequences of maladaptive central plasticity after nerve injury on motor function, rats were trained to perform an automated reach-and-grasp task that measured volitional forelimb strength (Fig. 1a; Supplementary Fig. 1; Supplementary Movie 1). Once proficient, the median and ulnar nerves in the trained forelimb were transected and a 6 mm guide conduit bridged the nerve stumps for each nerve[27] (Fig. 1b). This procedure results in total denervation of the muscles in the forelimb controlling digit flexion while sparing innervation of forelimb extensor muscles innervated by the radial nerve[27]. Reinnervation distal to the injury site occurs spontaneously, however, chronic weakness and deficits in nerve anatomy persist (Fig. 1b, Supplementary Figs. 10–13). Five weeks after injury, all rats underwent implantation of a stimulating cuff electrode on the left cervical branch of the vagus nerve in the neck (Fig. 1c). By the sixth week post-injury, rats could freely perform the reach-and-grasp task (Supplementary Movie 2). While regeneration continues for many months following nerve transection[28], return of skilled forelimb use indicates that substantial reinnervation had occurred[29,30]. Rats then began 7 weeks of daily rehabilitation, during which groups received either 0.5 s bursts of closed-loop VNS (CL-VNS) paired with volitional forelimb movements during rehabilitation, a matched amount of non-contingent VNS delivered 2 h after equivalent daily rehabilitation (Delayed VNS), or equivalent rehabilitation without VNS (Rehab; Fig. 1d). For all CL-VNS and Delayed VNS subjects, each 0.5 s VNS train consisted of 16 biphasic pulses of 0.8 mA and 100 μs pulse width delivered at 30 Hz. Previous studies demonstrate that this parameter set is optimal for driving VNS-dependent central plasticity[31–34].

Thirteen weeks after nerve injury and following the conclusion of rehabilitation, we assessed plasticity in the motor cortex using intracortical microstimulation[25]. Nerve damage generated a reduction in cortical area that evoked movements from the denervated digit flexors and an expansion of motor cortical area that evoked movements from the spared extensors (Fig. 2a; Uninjured vs. Rehab; Digit Flexion: $p = 5.02 \times 10^{-4}$; Extension: $p = 0.02$; unpaired $t$-test). These results are consistent with classical studies documenting the cortical areal reduction of injured circuits and a consequent expansion of spared circuits[7,8,35,36], and illustrate the long-term central network changes precipitated by peripheral nerve damage.

Pairing closed-loop VNS with training drives the reorganization of central motor networks specific to the trained movement[20,37]. Here, we tested if CL-VNS paired with reach-and-grasp training after nerve damage would restore the weakened digit flexion networks in motor cortex and subsequently reverse the cortical expansion of extension induced by nerve injury. CL-VNS significantly increased the area of motor cortex that evoked movements of the reinnervated digit flexors (Fig. 2a, b; Rehab vs. CL-VNS; $p = 0.0032$; unpaired $t$-test). In conjunction, we observed a reduction of the expanded cortical area evoking movements of the spared extensors (Fig. 2a, b; Supplementary Figs. 2–3, Supplementary Table 8; Rehab vs. CL-VNS; $p = 8.73 \times 10^{-4}$; Wilcoxon rank-sum). These findings suggest that CL-VNS reverses the maladaptive central reorganization resulting from nerve injury.

VNS-dependent plasticity relies on the precise timing of stimulation paired with training[25]. Thus, temporally decoupling VNS and rehabilitation should prevent VNS-dependent changes

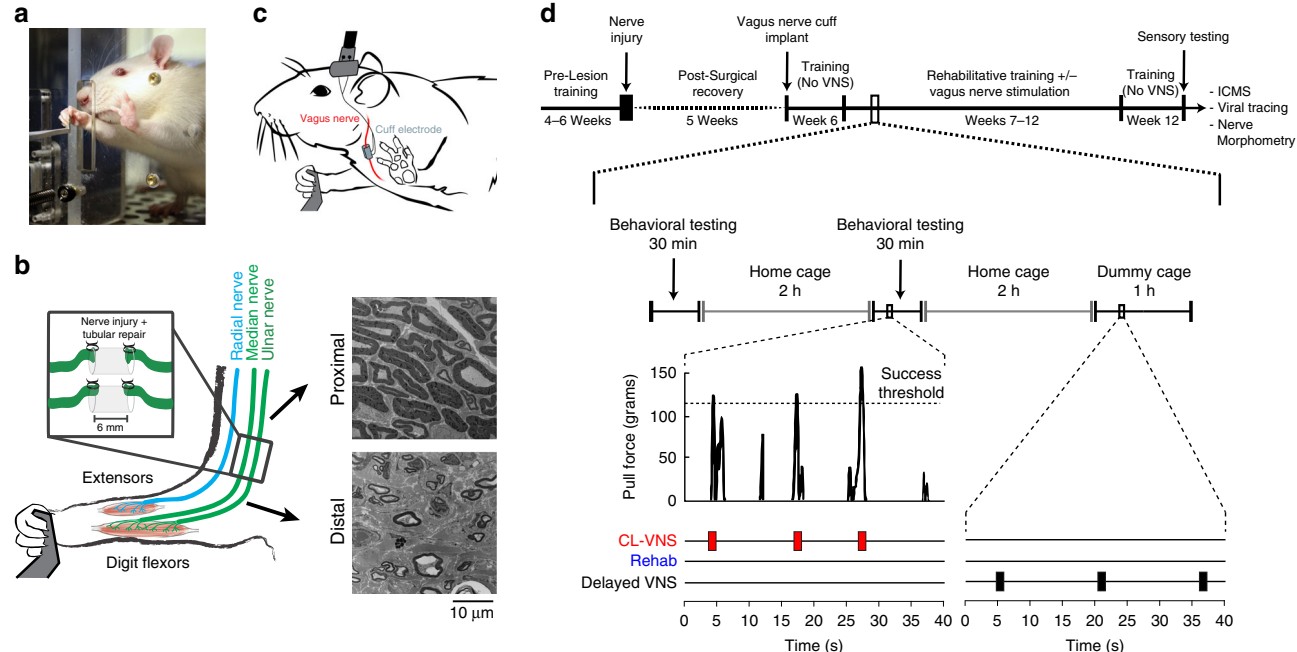

**Fig. 1 Median and ulnar nerve injury and delivery of vagus nerve stimulation during rehabilitation. a** A rat performing the volitional forelimb isometric pull task. **b** Schematic of the nerve injury. The median and ulnar nerves in the trained forelimb, innervating the digit flexor muscles required for grasping, were individually transected. The nerve stumps were then sutured into a guide conduit leaving a 6 mm gap between stumps. Reinnervation takes place, but the procedure results in chronic deficits in nerve architecture distal to the injury site. The radial nerve innervating the wrist and digit extensors of the forelimb was spared. Scale bar indicates 10 μm. **c** Illustration of the VNS device. A cuff electrode was placed on the left cervical vagus nerve in the neck with subcutaneous leads connected to a head-mount. **d** Timeline of rehabilitative training after nerve injury. Rehabilitation began 6 weeks after nerve injury, and rats received either closed-loop VNS paired with forelimb movement during rehabilitation (CL-VNS), equivalent training without VNS (Rehab), or a matched amount of VNS delivered after equivalent daily training sessions (Delayed VNS).

in central networks. Indeed, a matched amount of VNS (equivalent number of stimulation trains and identical VNS parameters) delayed after daily training failed to restore central networks (Fig. 2a, b; Delayed VNS vs. CL-VNS; Digit Flexion: $p = 0.0092$, Extension: $p = 9.32 \times 10^{-4}$; Wilcoxon rank-sum). This result indicates that the plasticity-enhancing effects of VNS depends on the precise timing of stimulation with rehabilitation.

Reinnervation after nerve transection invariably produces misdirected connectivity in which some axons form aberrant connections with multiple muscles[13,38]. This peripheral scrambling consequently results in the disorganization of central control networks, eroding selective recruitment of muscle groups[39,40]. Patients with nerve damage often develop debilitating and abnormal muscle synergies that greatly contribute to dysfunction[39,40]. Here, we observed that nerve damage led to the development of multi-joint movements involving simultaneous digit and elbow flexion in response to intracortical microstimulation (ICMS) (Fig. 2c, d; Uninjured vs. Rehab, $p = 4.54 \times 10^{-4}$, unpaired $t$-test). CL-VNS reversed this abnormal recruitment of simultaneous digit and elbow flexion in response to ICMS at equivalent stimulation intensities (Fig. 2c, d; Supplementary Fig. 4; CL-VNS v. Rehab, $p = 7.73 \times 10^{-4}$; Wilcoxon rank-sum). Delayed VNS, which does not augment central plasticity, failed to reverse this abnormal recruitment (Fig. 2c, d; CL-VNS v. Delayed VNS; $p = 0.011$; Wilcoxon rank-sum). Together, these findings demonstrate that delivery of CL-VNS drives central changes to modify the recruitment of forelimb musculature following nerve damage.

**CL-VNS enhances recovery after nerve damage.** Analysis of motor performance confirmed that enhancing central plasticity and reversing the maladaptive changes in central networks

produced by nerve damage improves recovery. Volitional fore-limb strength was substantially impaired in Rehab subjects even 12 weeks after nerve damage despite 7 weeks of intensive daily rehabilitation (Rehab; Fig. 3a, b; Supplementary Movie 3). In contrast, CL-VNS doubled recovery of motor function compared to rehabilitation alone (Fig. 3a, b; Supplementary Movie 4; Success Rate, Rehab v. CL-VNS; $p = 3.33 \times 10^{-7}$; Two-way RM ANOVA). Improved motor function persisted even after the cessation of stimulation during week 12, consistent with a restoration of network function. Delayed VNS failed to improve motor function, associating the absence of central plasticity with chronic motor dysfunction (Fig. 3a, b; Supplementary Figs. 5–8; Supplementary Tables 1–2; Success Rate, CL-VNS v. Delayed VNS; $p = 6.21 \times 10^{-6}$; Two-way RM ANOVA). Overall, the degree of motor recovery in rats that received CL-VNS paired with rehabilitative training was markedly improved, with all of the CL-VNS subjects displaying at least 80% recovery, whereas only 31% of Rehab and 36% of Delayed VNS subjects showed similar recovery (Fig. 3c; Recovery (%) = Pull Force $\left( \frac{\text{Week12} - \text{Week6}}{\text{Pre} - \text{Week6}} \right)$). Together, this evidence represents the first report that reversing maladaptive central plasticity can support the recovery of motor function after nerve damage.

Since numbness is a critical aspect of disability in many patients after nerve injury and is accompanied by changes in central networks[19], we reasoned that CL-VNS dependent reversal of maladaptive central plasticity may also produce improvements in tactile sensation. We tested tactile function 13 weeks post-injury, after considerable nerve regeneration had occurred. Nerve injury increased somatosensory thresholds of the forepaw, indicating a loss of sensation (Fig. 3d; Uninjured vs. Rehab, $p = 3.88 \times 10^{-4}$; unpaired $t$-test). CL-VNS improved sensory function compared to both Rehab and Delayed VNS groups, as

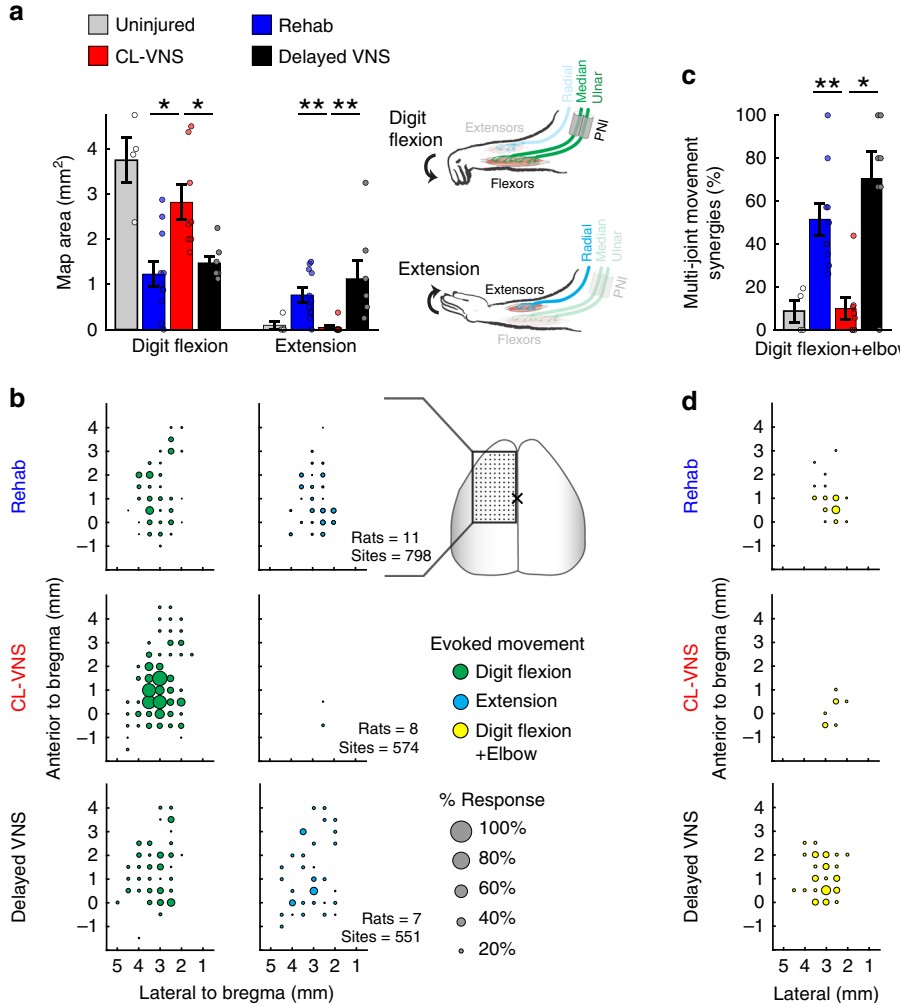

**Fig. 2 Closed-loop VNS restores cortical motor maps after nerve damage. a** Intracortical microstimulation reveals that, despite extensive rehabilitative training, nerve injury results in a substantial reduction in motor cortical area evoking movements of the denervated digit flexors and an increase in motor cortical area that evokes extension movements (Rehab, $n = 11$). CL-VNS ($n = 8$) reverses these lesion-induced cortical map changes, restoring the digit flexion representations and reducing the aberrant expansion of extensor representations. Delayed VNS ($n = 7$) failed to restore motor map representations, demonstrating that VNS relies on timed engagement with rehabilitation. **b** Bubble plots detailing the cortical locations of digit flexion and wrist extension movements across animals in each experimental group. The size of each bubble represents the proportion of subjects that the stimulation evoked a digit flexion or wrist extension movement at each cortical site. Note that CL-VNS significantly increases the denervated digit flexion representation and reduces the lesion-induced expansion of extension compared to Rehab and Delayed VNS. Same group sizes as in (**a**). **c** Due to misdirected reinnervation, nerve damage results in a substantial increase in the percentage of sites that generate multi-joint movements involving simultaneous digit and elbow flexion. CL-VNS reduced the percentage of sites that simultaneously elicit both movements compared to Rehab or Delayed VNS, suggesting the restoration of independent muscle control. Same group sizes as in **a**. **d** Bubble plots illustrating the stimulation site locations that generate the simultaneous digit and elbow flexion movements. Same group sizes as in **a**. Circles depict individual subjects in **a**, **c**. Error bars indicate S.E.M. All comparisons represent Bonferroni-corrected $t$-tests to CL-VNS, ***$p < 0.0005$, **$p < 0.005$, *$p < 0.025$. Source data are provided as a Source Data file.

demonstrated by a reduction in tactile threshold (Fig. 3d; Supplementary Fig. 9; Supplementary Tables 3–4; CL-VNS v. Rehab, $p = 0.009$; CL-VNS v. Delayed VNS, $p = 0.0085$; unpaired $t$-test). Thus, the enhanced recovery with closed-loop VNS indicates that enduring maladaptive plasticity contributes to persistent motor and sensory symptoms of nerve damage.

**CL-VNS does not improve peripheral nerve or muscle health.** We next sought to identify the anatomical changes that subserve recovery after nerve injury. To ascertain whether changes in the injured peripheral nerves could underlie recovery, we assessed multiple metrics of nerve health in the regenerated median and ulnar nerves and the reinnervated muscles at the conclusion of rehabilitation (week 13). No differences were observed in the size

or number of muscle fibers of the reinnervated forelimb muscles (Fig. 4a, b; Supplementary Fig. 14; Supplementary Table 7; $p > 0.05$; Kruskal–Wallis test). Additionally, electron microscopy morphometric analysis failed to show VNS-dependent differences in myelin thickness, number of myelinated axons, axon size, or G-ratio in the distal regenerated segments of the median or ulnar nerves (Fig. 4c, d; Supplementary Figs. 10–13; Supplementary Tables 5–6, 16–17; All $p > 0.05$, Kruskal–Wallis tests). These results suggest that CL-VNS does not improve recovery after nerve damage by improving nerve regeneration or muscle health.

**CL-VNS increases synaptic connectivity in central networks.** Based on the absence of anatomical differences in the periphery, we tested the hypothesis that CL-VNS paired with rehabilitation

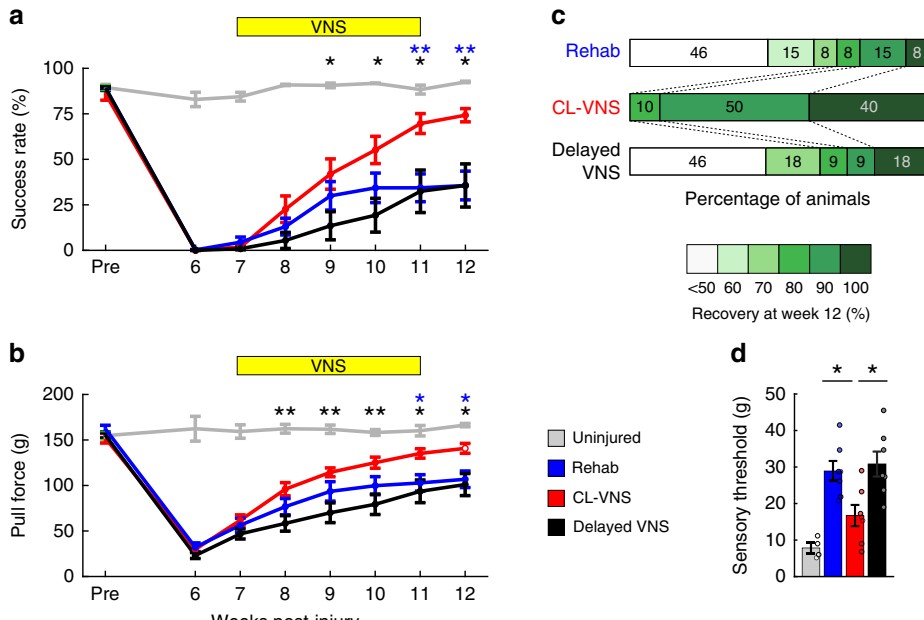

**Fig. 3 Closed-loop VNS delivered during rehabilitative training improves motor and sensory function after nerve damage. a**, **b** Closed-loop VNS paired with rehabilitation (CL-VNS, $n = 10$) significantly enhances recovery of volitional forelimb function compared to equivalent rehabilitation without CL-VNS (Rehab, $n = 13$). Consistent with the failure to restore cortical motor maps, Delayed VNS ($n = 11$) fails to improve recovery compared to CL-VNS. **c** Plots depicting the proportion of subjects in each group sorted by percentage of pull force recovery. All CL-VNS subjects recovered at least 80% of pre-injury function by the end of therapy. Subjects receiving either Rehab or Delayed VNS exhibited substantially reduced levels of recovery, with many subjects demonstrating < 50% recovery. Same group sizes as in **a**, **b**. **d** CL-VNS ($n = 7$) significantly improved tactile sensation in the denervated forepaw compared to both Rehab ($n = 7$) and Delayed VNS ($n = 7$) groups. In **a**, **b** asterisks indicate significant differences using $t$-tests across groups at each time point. The color of the asterisk denotes the group compared to CL-VNS (blue indicates CL-VNS v. Rehab, and black indicates CL-VNS v. Delayed VNS). Circles depict individual subjects in **d**. Error bars indicate S.E.M. All comparisons represent Bonferroni-corrected $t$-tests to CL-VNS, ***$p < 0.0005$, **$p < 0.005$, *$p < 0.025$. Source data are provided as a Source Data file.

would enhance anatomical connectivity within central networks. Following stroke and spinal cord injury, CL-VNS increases connectivity in corticospinal motor networks and this effect is associated with improved functional recovery[24,25]. Because CL-VNS increased cortical representations of the denervated digit flexion networks (Fig. 2a), we tested whether increased connectivity in corticospinal pathways could underlie this effect. We injected the eGFP-expressing retrograde transsynaptic tracer pseudorabies virus PRV-152 into the digit flexors innervated by the injured median and ulnar nerves to label central neurons synaptically coupled to the digit flexors. To investigate changes in central networks controlling extensor muscles innervated through the spared radial nerve, we injected RFP-expressing PRV-614 into the extensor carpi radialis longus (Supplementary Fig. 15). Consistent with the increase in cortical area evoking digit flexion of the forepaw in response to ICMS (Fig. 2a), CL-VNS increased the number of cortical neurons labeled by tracer injection into the forelimb digit flexors (Fig. 4e; Supplementary Table 9; $p = 0.017$; unpaired $t$-test). The increased number of cortical neurons coupled to digit flexion networks parallels the recovery observed on the reach-and-grasp task, suggesting that this increase may contribute to functional improvements. No differences were observed in the number of labeled cortical neurons after injection into the extensor muscles innervated by the spared radial nerve (Fig. 4f; $p = 0.817$; unpaired $t$-test). Because CL-VNS reduced the abnormal co-recruitment of digit flexion—elbow flexion movements resulting from nerve injury (Fig. 2c), we investigated whether an anatomical correlate could explain this finding. CL-VNS reduced the percentage of cortical neurons co-labeled by digit flexor and extensor muscle injections (Fig. 4g; $p = 0.0013$; unpaired $t$-test). These findings provide evidence that CL-VNS

delivered with rehabilitation drives robust anatomical reorganization in central networks after nerve damage.

**Blocking cortical plasticity prevents CL-VNS recovery**. Despite the association of central plasticity and functional recovery after nerve damage, the experiments above do not establish a causal relationship between these phenomena. Therefore, we performed a second set of experiments to directly test the hypothesis that augmenting central plasticity is required for improvements in motor and sensory function after nerve injury. If these observations are causally linked, then selectively blocking the reversal of maladaptive plasticity in CL-VNS subjects should prevent improved recovery. To test this, we evaluated plasticity and recovery after nerve damage in a group of rats treated with a cell-type specific immunotoxin to deplete acetylcholine in the brain, a neuromodulator known to be required for CL-VNS-dependent plasticity[20]. Because acetylcholine depletion is restricted to cortical circuits, any differences between ACh-:CL-VNS and CL-VNS subjects can be ascribed to central effects of closed-loop VNS.

Depletion of acetylcholine blocked the CL-VNS-dependent reversal of lesion-induced plasticity in the cortex (Fig. 5a; Supplementary Tables 12–13). After CL-VNS therapy in acetylcholine-depleted rats (ACh-:CL-VNS), we did not observe any sites in motor cortex that generated movements of the reinnervated digit flexors, compared to a restoration of cortical digit flexion representations in CL-VNS rats with normal cholinergic signaling (Fig. 5a, b; CL-VNS v. ACh-:CL-VNS; $p = 0.028$; Mann–Whitney $U$). Furthermore, a substantial portion of motor cortex in acetylcholine-depleted rats (ACh-:CL-VNS)

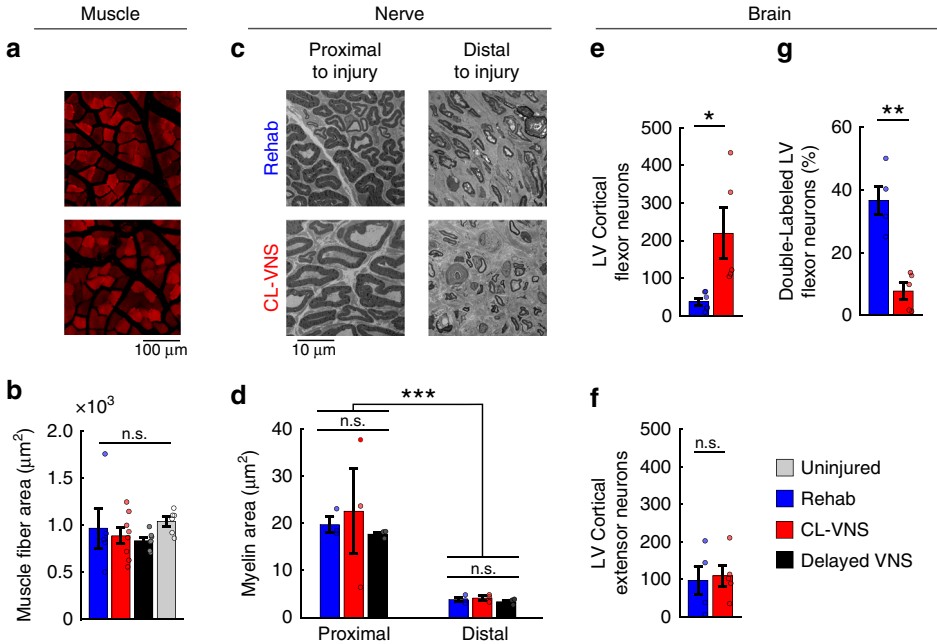

**Fig. 4 Closed-loop VNS increases putative central synaptic connectivity of injured networks without altering peripheral reinnervation. a** Representative images depicting muscle fibers from the denervated digit flexors. Muscle fibers were visualized using non-specific background fluorescence. Scale bar is 100 μm. **b** No differences were observed in fiber area or other metrics of muscle morphology (Rehab, $n = 5$; CL-VNS, $n = 8$; Delayed VNS, $n = 7$; uninjured, $n = 6$; Kruskal–Wallis test, see Supplementary Fig. 14). Histology on nerve and muscle sections was collected from subjects at week 13. **c** Representative images of nerves harvested from subjects at the conclusion of rehabilitation illustrating nerve morphology proximal and distal to the injury site in both a Rehab and CL-VNS subject. The distal sections show some reinnervation, but substantially impaired nerve architecture in subjects from all groups. Scale bar is 10 μm. **d** Nerve injury substantially disrupts nerve architecture distal to the injury site in the median nerve (Rehab, $n = 3$; CL-VNS, $n = 3$; Delayed VNS, $n = 3$, Kruskal–Wallis test). However, no differences in myelin area or other measures of nerve health were observed across groups (see Supplementary Figs. 10–13), indicating that VNS does not influence peripheral reinnervation. **e** Pseudorabies virus (PRV) tracing of putative central synaptic connectivity reveals that CL-VNS delivered with rehabilitation ($n = 5$) significantly increased the number of neurons in layer 5 (LV) motor cortex synaptically coupled to forelimb digit flexors compared to Rehab ($n = 5$), consistent with the expansion of digit flexion movement representations in the cortical maps. Same group sizes in **e–g**. **f** No differences were observed in the connectivity of the spared extensor networks. **g** CL-VNS significantly reduced the percentage of neurons displaying connectivity to both digit flexion and extensor muscles. Circles depict individual subjects in **b**, **d–g**. Error bars indicate S.E.M. All comparisons in **e–g** represent unpaired $t$-tests, ***$p < 0.001$, **$p < 0.01$, *$p < 0.05$. Source data are provided as a Source Data file.

elicited movements of the spared extensors, compared to the absence of this aberrant movement representation in CL-VNS subjects (Fig. 5a, b; Supplementary Fig. 16; CL-VNS v. ACh-:CL-VNS; $p = 0.028$; Mann–Whitney $U$ test).

Because depletion of acetylcholine blocked reversal of the lesion-induced central plasticity after nerve injury, we expected that it would also prevent improved recovery of motor and sensory function after CL-VNS therapy. Indeed, cortical cholinergic depletion blocked enhanced recovery of motor function observed in CL-VNS rats with intact cortical cholinergic innervation (Fig. 5c, d; Supplementary Figs. 17–19; Supplementary Tables 10–11; Success Rate, CL-VNS v. ACh-:CL-VNS; $p = 3.53 \times 10^{-10}$; Two-Way RM ANOVA). Similarly, depletion of acetylcholine prevented CL-VNS-dependent improvement of forelimb tactile function (Fig. 5e; Supplementary Fig. 20; Supplementary Tables 14–15; CL-VNS v. ACh-:CL-VNS; $p = 1.26 \times 10^{-3}$; unpaired $t$-test), suggesting that cortical circuits facilitate tactile improvements after CL-VNS therapy. In summary, these results demonstrate that reorganization exclusively in central networks can reverse the persistent motor and sensory deficits arising from nerve damage.

## Discussion
Damage to the peripheral nervous system is a major source of disability for millions around the world. The majority of therapy development is directed at reparative and regenerative strategies

to restore peripheral connectivity, with other therapeutic avenues largely unexplored[5]. In this study, we provide compelling evidence that peripheral connectivity is not the sole limiting factor for recovery, and that maladaptive central changes in response to nerve injury contribute to dysfunction. Thus, targeting maladaptive central plasticity represents a therapeutic strategy for nerve injury. We show that reversing the maladaptive central plasticity arising from nerve damage is sufficient to improve motor and sensory function. These benefits were subserved by synaptic reorganization in central circuits controlling the reinnervated muscles and occurred in the absence of changes in peripheral connectivity. Finally, preventing the plasticity-enhancing effects of CL-VNS by either temporally decoupling VNS from rehabilitation or specifically depleting acetylcholine in the brain blocked the improved recovery of CL-VNS subjects, providing a causal link between insufficient central plasticity and motor and sensory deficits generated by peripheral nerve damage.

The central and peripheral nervous systems are functionally integrated, and disruption of peripheral signaling after nerve injury results in profound reorganization of central networks[8–11,14,41]. In conjunction with the well-characterized structural deficiencies in reinnervated fibers, central changes are hypothesized to be maladaptive and contribute to dysfunction[5]. Several lines of indirect evidence support the role of the central nervous system in recovery after nerve injury. A study in rats demonstrated that promoting plasticity in spinal networks improved functional recovery after

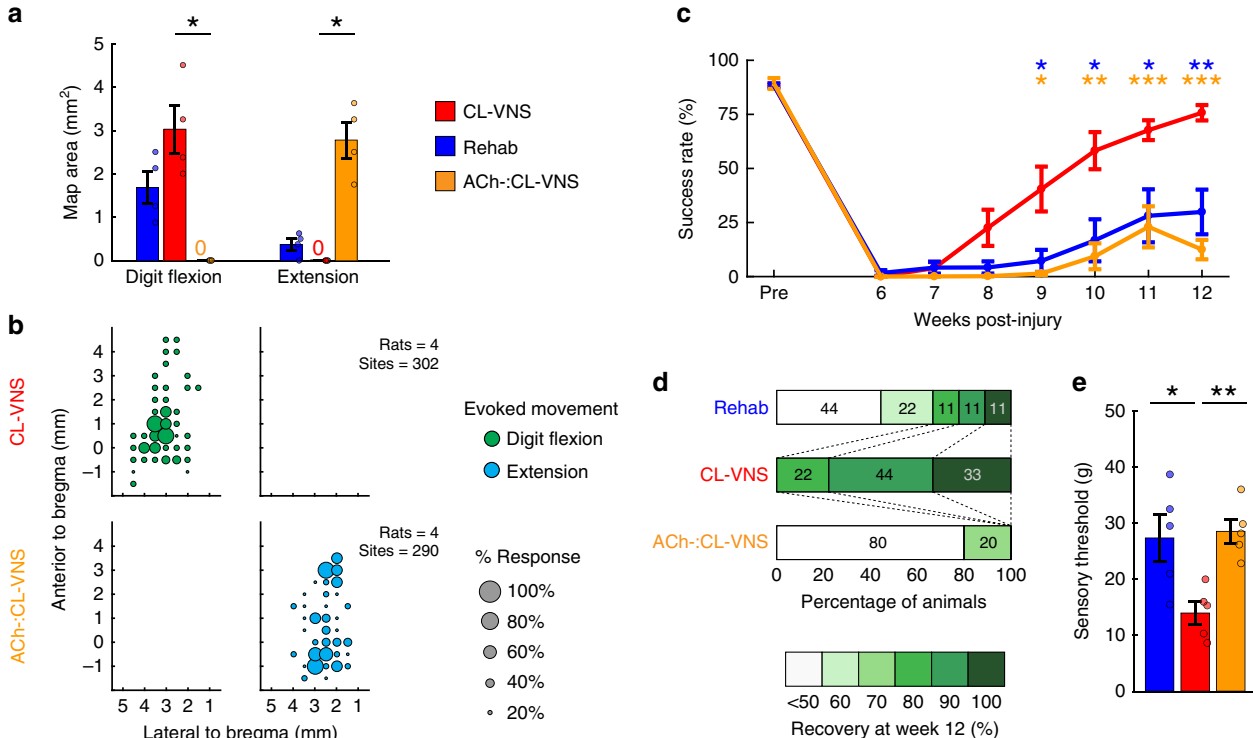

**Fig. 5 Depleting acetylcholine in the brain prevents the reversal of pathological plasticity and blocks motor and sensory recovery after nerve injury. a** Depletion of acetylcholine in rats that receive equivalent VNS paired with rehabilitation (ACh-:CL-VNS, $n = 4$) blocks the VNS-dependent restoration area of motor cortex evoking movements of the denervated digit flexors (CL-VNS, $n = 4$; Rehab, $n = 4$). Additionally, CL-VNS reverses the aberrant expansion of evoked extension movements, while acetylcholine-depleted subjects display an enduring expansion of extension representations. Compare to Fig. 2. **b** Bubble plots displaying the cortical locations of digit flexion and extension movements across all animals in each experimental group. The size of each bubble represents the proportion of subjects for which ICMS evoked digit flexion or extension movements at each cortical site. Note that depletion of acetylcholine prevents CL-VNS-dependent restoration of digit flexion representations and also prevents the reduction of extensor representations. Same group sizes as in **a**. **c** Consistent with blocking the reversal of pathological plasticity, depletion of acetylcholine prevents the VNS-dependent enhancement of recovery of motor function after nerve injury (Rehab, $n = 9$; CL-VNS, $n = 9$; ACh-:CL-VNS, $n = 5$). **d** ACh-:CL-VNS subjects exhibit substantially reduced levels of motor recovery compared to CL-VNS. Same group sizes as in **c**. **e** Depletion of acetylcholine also prevents VNS-dependent recovery of tactile thresholds (Rehab: $n = 5$; CL-VNS: $n = 5$; ACh-:CL-VNS: $n = 5$). Circles depict individual subjects in panels (**a**, **e**). In **c** asterisks indicate significant differences using t-tests across groups at each time point, and the color of the asterisk denotes the group compared to CL-VNS. All comparisons represent Bonferroni-corrected t-tests to CL-VNS, ***$p < 0.0005$, **$p < 0.005$, *$p < 0.025$. Source data are provided as a Source Data file.

nerve injury[42]. Furthermore, previous studies in humans have demonstrated associations between sensory recovery and brain structure and cortical processes[5,16,18,19]. Finally, functional outcomes after nerve repair are remarkably better in children despite a similar degree of reinnervation inaccuracies, an effect ascribed to heightened plasticity[43]. Building on this logic, we implemented a technique known to drive robust and long-lasting central plasticity to directly test whether enhancing reorganization in central networks could promote recovery after nerve injury. Here, we observed long-lasting alterations in the cortical motor maps, including reductions in digit flexion and an expansion of extension representations, an effect characterized for the first time in the forelimb of the rat. Closed-loop VNS reversed these alterations and improved motor function, suggesting that the map changes observed are pathological and likely limit recovery. In addition to reversing the maladaptive plasticity arising from nerve injury, subjects also displayed enhanced recovery of motor and sensory function without any observable changes in the peripheral nerves or muscles (Fig. 6a–c). These findings provide direct evidence that insufficient central plasticity can interfere with functional recovery after nerve damage and demonstrates that techniques that reverse maladaptive plasticity that occurs after nerve damage hold promise for improving function in the chronic phase of injury. Importantly, this strategy acts in conjunction with target reinnervation, and thus

could be employed synergistically with therapeutics that aim to promote peripheral regeneration.

VNS exerts a broad range of central and peripheral actions, including parasympathetic activation[44] and regulation of the immune system[45], which could influence peripheral reinnervation and recovery independent of central plasticity. To assess the contribution of plasticity-independent effects of VNS on recovery, the Delayed VNS and ACh-:CL-VNS control groups were incorporated in to the study design. Ascending fibers in the cervical vagus nerve synapse bilaterally on neurons within the nucleus tractus solitarius in the brainstem, which subsequently project to neuromodulatory centers including the cholinergic basal forebrain[46–49]. A large body of evidence demonstrates that CL-VNS enhances central plasticity by the temporally precise engagement of neuromodulatory networks during rehabilitative exercises[22]. In addition to the CL-VNS pairings utilized in this study, growing evidence demonstrates that brain and spinal stimulation facilitates recovery when precisely paired with rehabilitation, indicating that neurostimulation techniques rely on temporal precision with ongoing neural activity to reorganize central neural networks to improve recovery[50,51]. To investigate the contribution of VNS effects that do not require this precise closed-loop timing, we assessed plasticity and recovery in the Delayed VNS subjects who received a matched amount of

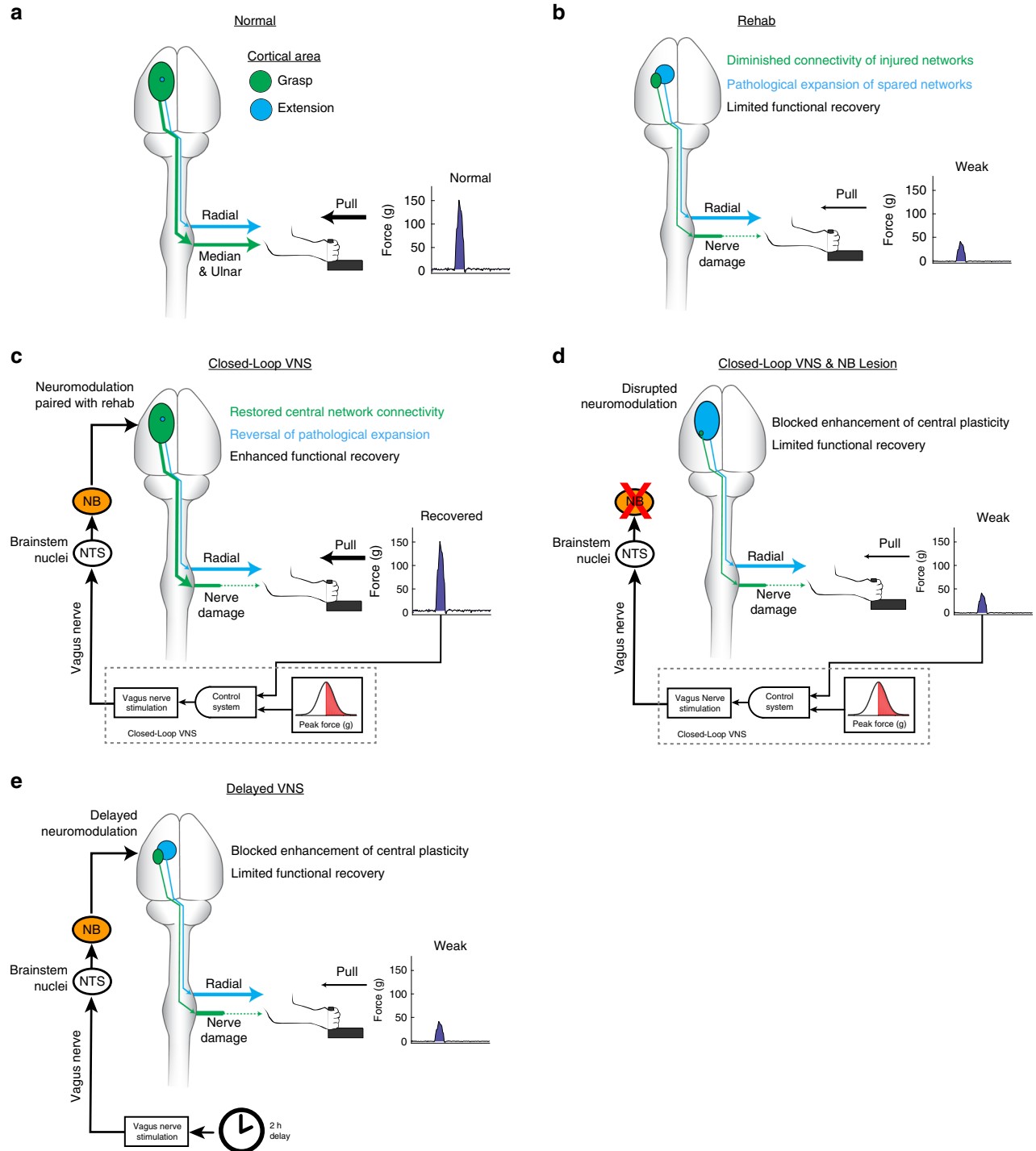

VNS but stimulation was delivered 2 h after equivalent daily rehabilitative training. Any timing-independent effects of VNS, such as modulation of the immune system or on regeneration, would likely still occur in these subjects. Consistent with previous studies, Delayed VNS failed to enhance plasticity and prevented improved recovery. Analysis of morphological measures of the injured nerve and denervated muscles did not reveal any VNS-dependent effects on nerve or muscle health, suggesting that the beneficial effects of CL-VNS do not act through improving peripheral health. This result demonstrates that VNS alone is insufficient for improved recovery after nerve damage, and CL-VNS enables plasticity and improves recovery by delivering

temporally precise engagement of neuromodulatory networks during rehabilitative training (Fig. 6c, e).

To investigate a causal link between enhancement of central plasticity and improved recovery after nerve damage, a subset of CL-VNS subjects received lesions that depleted cortical acetylcholine specific to the forebrain (Fig. 6d). Acetylcholine is known to be involved in cortical plasticity[52–54], and depletion of cortical acetylcholine blocks CL-VNS enabled plasticity[20]. Thus, any effects observed in CL-VNS subjects but not in ACh-:CL-VNS subjects can reasonably be ascribed to actions that require acetylcholine in the brain. As expected, depletion of acetylcholine blocked CL-VNS enabled plasticity after nerve injury. In

**Fig. 6 Proposed mechanism underlying VNS-mediated enhanced plasticity and recovery after nerve damage. a** Prior to injury, the majority of motor cortex evokes movements of the digit flexors through the median and ulnar nerves, and a small area evokes movements of the extensor muscles through the radial nerve. Subjects are able to grasp and pull a handle to generate around 150 g of force. **b** Damage to the median and ulnar nerves generates maladaptive changes in central networks. Cortical drive and synaptic connectivity within the injured digit flexion networks is reduced. In conjunction, networks innervated through the spared radial nerve demonstrate a pathological cortical expansion. Despite reinnervation and rehabilitative training, subjects continue to exhibit maladaptive central changes and long-term deficits in force generation. **c** Closed-loop VNS paired with forelimb movement during rehabilitation generates timed activation of neuromodulatory networks, including the cholinergic nucleus basalis (NB). This precisely timed neuromodulation enhances synaptic connectivity and cortical drive to increase output to muscles via the reinnervated median and ulnar nerves. Correspondingly, CL-VNS reverses the pathological expansion of extensor networks controlled via the spared radial nerve. Motor function is recovered in the absence of large scale peripheral changes in the nerves or muscle, indicating that central changes can compensate and restore function. **d** Lesion of the nucleus basalis (NB) prevents acetylcholine release and consequently CL-VNS-dependent central plasticity. No reorganization of central networks was observed in subjects with NB lesions that received CL-VNS. NB lesions prevented the enhanced recovery seen with CL-VNS, providing a causal link between enhanced central reorganization and improved recovery after nerve damage. **e** CL-VNS is based on precise timing between the activation of neuromodulatory networks and neural activity during rehabilitation. A matched amount of VNS delayed by 2 h from equivalent rehabilitation degrades the temporal association and prevents CL-VNS-dependent plasticity in central networks. Consequently, subjects that receive Delayed VNS fail to demonstrate enhanced recovery of function. This illustrates that timing-independent effects of VNS cannot account for enhanced recovery, and reinforces the notion that plasticity in central networks directed by CL-VNS supports the recovery of motor and sensory function after nerve damage.

conjunction, ACh-:CL-VNS subjects failed to display enhanced recovery of motor or sensory function, consistent with the notion that central plasticity is associated with recovery after nerve injury. Because depletion of acetylcholine was restricted to cortical networks, it is unlikely that CL-VNS improves recovery by affecting nerve regeneration. The requirement for acetylcholine in recovery also has important clinical implications, as concurrent treatment with pharmaceuticals or comorbidities that affect central cholinergic transmission may influence efficacy of CL-VNS therapy. While acetylcholine was the primary focus of the present study, multiple other neurotransmitters regulate plasticity resulting from nerve damage[55]. VNS modulates neural activity in the noradrenergic locus coeruleus (LC) and the serotonergic dorsal raphe nucleus (DRN)[21,56], two neuromodulatory centers that intimately regulate cortical plasticity. VNS-dependent enhancement of plasticity likely relies on engagement of multiple neuromodulatory networks[22,57]. Together, these results support a mechanism by which closed-loop VNS provides precisely timed release of acetylcholine during rehabilitative training to enhance plasticity in motor networks to improve recovery (Fig. 6). Additionally, these findings provide direct support for a causal link between central plasticity and recovery after nerve damage.

Analysis of the reorganization in central networks after nerve injury provides insight into the mechanisms that likely subserve CL-VNS-dependent recovery. Nerve injury generates profound reorganization throughout central networks that persist even after reinnervation[5]. Here, nerve injury generated a substantial increase in the area of motor cortex producing wrist extension, a movement innervated by the spared radial nerve. Our results support that this aberrant expansion of spared networks is maladaptive, and these findings corroborate the effects documented in classical studies[5,36]. These maladaptive changes are believed to arise from the transient loss of signaling of the denervated network, allowing spared network activity to dominate. The resulting overrepresentation of the spared network persists even after peripheral connectivity in the injured network is regained, due to the inability of the weakened network to compete with the dominant synaptic activity of the spared network. If maladaptive plasticity directly contributes to dysfunction after nerve injury, manipulations that enhance potentiation in the weakened reinnervated networks or enhance depression in spared networks may allow reversal of maladaptive plasticity and consequently lead to recovery.

Previous studies document the ability of CL-VNS to enhance plasticity in motor networks specific to the paired movement[20,37].

Here, we observed that CL-VNS delivered with reach-and-grasp rehabilitation substantially increased the cortical area generating movements of reinnervated digit flexion networks (Fig. 2a). This CL-VNS-dependent expansion of cortical digit flexion area revealed in the ICMS studies was associated with an increase in connectivity of the corresponding digit flexion central networks, likely in corticospinal connections[24,58] (Fig. 4e). In conjunction with the potentiation of denervated networks, CL-VNS significantly reduced the maladaptive expansion of the spared extensor networks (Fig. 2a). Notably, no changes were observed in extensor corticospinal connectivity (Fig. 4f). This result is consistent with the notion that cortical overrepresentation of spared networks after nerve injury results from the synaptic revealing of latent intracortical connections rather than peripheral connectivity changes[59].

Misdirection of regenerating axons after nerve transection produces aberrant peripheral connectivity in which regenerated axons establish connections with muscles not previously innervated, or even multiple muscles simultaneously. These aberrant connections lead to the development of debilitating synergies that degrade isolated muscle control. Here, we observed that nerve transection produced abnormal muscle recruitment involving digit flexion and elbow flexion movements (Fig. 2c). While the development of these abnormal muscle recruitments, or synergies, likely arises from misdirected peripheral connectivity, the elimination of this synergy with CL-VNS suggests that central networks can compensate for misdirected reinnervation through central synaptic depression. The failure of Delayed VNS to reduce this abnormal synergy provides additional evidence that the synergy elimination with CL-VNS is centrally mediated. Furthermore, CL-VNS reduced a putative anatomical correlate of muscle synergies, as defined by the number of co-labeled cortical neurons after tracing from the digit flexion and extensor muscles of the forelimb (Fig. 4g). We reasoned that co-labeled cortical neurons represent aberrant central synaptic connectivity, as the distance between the two muscles injected with virus are likely beyond the spatial limitations of collateral sprouting[60,61] and muscle health is relatively normal (Fig. 4b). CL-VNS therapy reduced the number of co-labeled cortical neurons, mirroring the elimination of the abnormal muscle synergy observed in the cortical maps and providing anatomical evidence of improved selectivity of motor networks after CL-VNS therapy. Together, the plasticity-enhancing effects of CL-VNS, the lack of peripheral changes, and the absence of recovery in the VNS control groups led us to conclude that CL-VNS reverses the aberrant overlap of central motor networks after nerve damage. These findings

support a mechanism by which CL-VNS delivered with rehabilitation enhances reorganization in corticospinal networks to enhance motor selectivity and improve recovery of motor function.

The present study provides a proof-of-concept demonstration of closed-loop VNS (CL-VNS) delivered with rehabilitation as a strategy to improve motor and sensory function after nerve injury, but some limitations merit consideration. Pain and dysfunction in temperature sensation are a common consequence following nerve damage and were not evaluated in the present study. Maladaptive central changes are associated with the percept of pain, thus CL-VNS paired with an appropriate sensory rehabilitation regimen address this symptom[22,62]. Although the preclinical assessments were designed to best capture aspects of dysfunction to provide a strong translational rationale, the assessment of skilled forelimb function and mechanical sensory withdrawal thresholds fail to capture the full complexity of motor control and sensation. Future studies that provide a more comprehensive characterization of motor and sensory function may be useful in further developing CL-VNS therapy to target additional aspects of dysfunction. Emerging evidence demonstrates that the parameters of CL-VNS influence the degree of plasticity in the central nervous system[31,32,34,63,64]. While the present study utilized the most common implementation of closed-loop VNS, continued optimization of stimulation paradigms in future studies may improve the clinical utility of CL-VNS therapy[21,25,31,32,34,63,64]. The experiments investigating nerve morphology after CL-VNS therapy had a relatively low sample size and substantial variability. Future studies should more extensively characterize peripheral health after VNS, including any potential effects of VNS on nerve physiology and axonal pathfinding. If, in addition to central modifications, CL-VNS acts through peripheral mechanisms not assessed in the present study, potential therapeutic regimens might incorporate VNS soon following injury to augment regeneration and further improve functional outcomes. Lastly, the immunotoxin used in the present study (192 IgG-Saporin) is known to affect other systems, including reductions of GABAergic markers and arc protein expression[65]. Thus, the present study cannot rule out the role of these systems in VNS-dependent recovery.

Pairing VNS with rehabilitation has emerged as a clinically viable therapy to treat a wide range of neurological disorders[25,37,66]. Evidence from preclinical models demonstrates that CL-VNS delivered with rehabilitative training enhances plasticity and promotes recovery in mechanistically distinct models of CNS damage, including stroke, intracerebral hemorrhage, spinal cord injury, and traumatic brain injury[24,25,67]. Highlighting the clinical potential of this strategy, two recent clinical trials demonstrate that CL-VNS delivered with physical rehabilitation enhances recovery of upper-limb function in chronic stroke patients[23,26]. A multimodal approach involving the integration of regenerative strategies to improve reinnervation combined with CL-VNS therapy may provide a unique approach to improving dysfunction following nerve damage. Coupled with a long track recovery of safety, this study positions CL-VNS delivered with rehabilitation as a powerful, readily translatable strategy to bring relief to millions with a range of disorders related to nerve damage.

## Methods

**Subjects**. 106 adult female Sprague-Dawley rats were studied, each weighing ~250 g when they entered the study. Female rats were used due to ease of handling, and because the behavioral measures[25], lesion model[29], and VNS parameters[24,25,63,64] have all been extensively optimized in this sex. All rats were maintained above 85% of their ideal body weight for age. The rats were housed in a 12:12 reversed light cycle environment, and behavioral training was performed during the dark cycle to increase daytime activity levels. This study complies with all relevant ethical

regulations for animal testing and research. All handling, housing, surgical procedures, and behavioral training were approved by the University of Texas at Dallas Institutional Animal Care and Use Committee.

**Behavioral apparatus and software**. The behavioral chamber consists of a clear acrylic cage (30 cm × 13 cm × 25 cm) with a 1 cm wide slot on the right edge of the front wall (MotoTrak Base Cage, Vulintus, Inc., Dallas, TX)[27]. The slot restricts use to the right forelimb while allowing full range of movement during interaction with the device. An aluminum pull handle was centered in the slot at a height of 6 cm from the cage floor. The pull handle was mounted on a metal slide which allowed the device to be placed at various fixed distances relative to the inside wall of the cage (Pull Behavior Module, Vulintus, Inc., Dallas, TX). A force transducer measured the force applied to the pull handle with a resolution of 0.1 g. Custom MATLAB software was used to control the task. A microcontroller sampled the force transducer at a frequency of 100 Hz, and the signal was passed to the computer for data display, control of behavioral sessions, and data storage for analysis.

**Isometric pull behavioral testing**. Continuous force transducer data was collected and stored on a trial-by-trial basis for each animal[25]. Trial initiation occurred when the animal generated 10 g of force on the handle. Animals were required to exceed the pre-determined force threshold within 2 s of trial initiation to receive a reward pellet and record a successful trial (Fig. 1d). If the pull force did not exceed the threshold within 2 s, the trial was recorded as a failure, and no reward pellet was given. Each trial was followed by a 2 s timeout, during which a trial could not be initiated. All activity 1 s prior to and 4 s following trial initiation was recorded for analysis. Reward pellets (45 mg dustless chocolate precision pellet, BioServ, Frenchtown, NJ) were delivered from pellet dispensers (Pellet Dispenser, Vulintus Inc., Dallas, TX) upon successful completion of a trial.

Animals underwent two 30-min behavioral training sessions daily, 5 days per week, with at least a 2-h interval between training periods. During the initial phases of training, the pull handle was placed 0.5 inches inside the cage wall, and the reward threshold was set to 10 g. An experimenter encouraged animal interaction with the handle using ground pellet dust. When the animal began to interact with the handle independently, the handle was retracted outside the cage in 0.25 inch increments to a final location of 0.75 inches outside relative to the inner cage wall. After that, behavioral testing continued using an adaptive thresholding program.

The program used the median of the peak pull force of the immediately preceding 10 trials to calculate the current trial threshold, with programmable minimum and maximum adaptive threshold bounds[27]. Using this algorithm, the threshold was progressively scaled throughout a behavioral session based on performance. All animals in this study trained with a reward threshold minimum of 10 grams and a maximum of 120 g (i.e., the success threshold for any trial was never less than 10 g or greater than 120 g). Success rate in this study was defined as the percentage of trials greater than the maximum threshold of 120 g. Training continued until animals achieved a ≥85% success rate averaged across 10 consecutive training sessions. Data from the 10 sessions was used for the "Pre" time point in all analyses. At this point, animals underwent the peripheral nerve injury surgical procedure.

No behavioral testing was performed for 5 weeks following nerve injury to allow for nerve reinnervation, similar to previous studies[27,30]. For one week following the VNS cuff implant surgery, animals remained in their home cage and did not perform the task. Behavioral testing then continued twice daily for 7 weeks (Fig. 1d) and animals were allowed to freely perform the task. Analyses were split into 1 week epochs, with each weekly time point consisting of 10 consecutive sessions (2 sessions per day for 5 weekdays).

**Peripheral nerve lesions**. Peripheral nerve injuries were performed within 1 week of reaching proficiency on the isometric pull task. The median and ulnar nerves proximal to the elbow were completely transected and repaired using a saline-filled tube[27]. Animals were anesthetized with ketamine hydrochloride (80 mg/kg, i.p.) and xylazine (10 mg/k, i.p.), and given supplemental doses as needed to maintain anesthesia levels. A small incision on the forelimb proximal from the elbow was made, and the median and ulnar nerves were carefully isolated and exposed. Each nerve was completely transected 1 cm proximal to the elbow. Immediately following transection, the proximal and distal stumps of each nerve were sutured 1 mm from the ends of a 8 mm saline-filled polyurethane tube (Micro-Renathane 0.095" I.D 0.066" O.D., Braintree Scientific, Inc., Braintree, MA), resulting in a 6 mm gap between nerve stumps. The skin incision was sutured and treated with antibiotic ointment. All animals were given baytril (7.5 mg/kg) immediately following surgery and sustained release buprenorphine (1.2 mg/kg) for 6 days following injury. All animals were placed in Elizabethan collars for 14 days following injury to prevent excessive grooming and autophagia of the denervated limb. Two animals in this study displayed autophagia in the first 2 weeks following PNI and were removed from the study. No animals exhibited autophagia following the 2 week period after injury. Animals were given 5 weeks of recovery before the cortical cholinergic depletion and/or vagus nerve cuff implant surgeries were performed.

**Vagus nerve cuff implant and stimulation delivery**. All subjects underwent implantation of a VNS stimulating cuff and head-mounted connector 5 weeks after nerve injury[24]. A four-channel connector was attached to four bone screws placed in the skull above the cerebellum and surrounding the lambdoid sutures. The left cervical branch of the vagus nerve was exposed after careful blunt dissection of muscle and surrounding fascia. A stimulating cuff was placed around the vagus nerve and leads from the cuff were tunneled subcutaneously and attached to the connector on top of the head. Exposed leads were covered in acrylic and incisions on the head and neck were sutured. A transient drop in blood oxygen saturation in response to a short (~2 s) VNS train was used to confirm that the cuff electrode and headcap was functional.

During rehabilitative training sessions in the behavioral chamber, a cable attached to a slip ring was plugged in to the headcap for the course of rehabilitation. The software monitoring and collecting the data signal from the force transducer provided a trigger signal to the isolated pulse constant current stimulator (AM Systems, Model 2100 Isolated Pulse Stimulator, Sequim, WA). For subjects in the CL-VNS and ACh-:CL-VNS groups, the trigger was sent to the stimulator to deliver VNS immediately when the pull force crossed the adaptively scaled success threshold (median of previous 10 trials; minimum threshold: 10 g; maximum threshold: 120 g) (Fig. 1d). The Delayed VNS group received 1 h of VNS in a separate chamber 2 h following daily behavioral training. Stimulation was triggered every 16 s, resulting in a total of 225 stimulations per day to match the amount of stimulation delivered to CL-VNS subjects. For all subjects receiving VNS (CL-VNS, Delayed VNS, and ACh:CL-VNS groups), each 500 ms train of VNS consisted of 16 0.8 mA 100 μs biphasic pulses delivered at 30 Hz. Previous studies to evaluate stimulation parameters have demonstrated that the parameters chosen here for closed-loop VNS are optimal for augmenting central plasticity[23–25]. Stimulation was delivered in the appropriate groups during weeks 7–11 (Fig. 1d). No VNS was delivered in any group on week 12 to examine effects lasting after the cessation of stimulation (Fig. 1d). VNS cuff impedance was monitored daily and animals were removed from the study if cuff impedance exceeded 15 KOhms. To evaluate activation of the vagus nerve at the conclusion of the study, we measured rapid stimulation-dependent depression of blood oxygen saturation, an effect ascribed to the Hering-Breuer reflex[68] (Supplementary Table 22). Animals were excluded from the study if no blood oxygen saturation drop was observed following 10 s trains of VNS (0.8 mA, 100 us biphasic pulses, 30 Hz)[68].

**Cortical cholinergic depletion**. A subset of animals underwent injections of an immunotoxin to deplete cortical acetylcholine prior to receiving CL-VNS therapy[20]. A subset of rats ($N = 8$) were anesthetized with ketamine hydrochloride (80 mg/kg, i.p.) and xylazine (10 mg/k, i.p.), and given supplemental doses as needed to maintain anesthesia levels. Rats were placed in a stereotaxic frame (David Kopf Instruments, Tujunga, CA) and burr holes were drilled over the nucleus basalis bilaterally. Rats received injections of either conjugated 192-IgG-Saporin (ACh-:CL-VNS, $N = 5$; ACh-:Rehab, $N = 4$) (Advanced Targeting Systems, San Diego, CA) to selectively lesion cholinergic neurons in the basal forebrain[69], or control injections of an untargeted antibody and saporin (ACh+:CL-VNS, $N = 3$), which does not enter cells and induce cell death. Toxin or control peptide (0.375 mg/mL in saline) was injected through a syringe and 32 gauge needle (1.0 μL Neuros Model 7001, Hamilton). Injections were made at the following sites (site 1&2: 0.3 μL, AP: −1.4, ML: ± 2.5, DV: −8.0; sites 3&4: 0.2 μL, AP: −2.6, ML: ± 4, DV: −7.0) at a rate of 0.025 μL every 15 s. The needle remained in place for 5 min following each injection to prevent backflow and allow diffusion of the toxin. Following injections, burr holes were filled with Kwik-cast Sealant (World Precision Instruments, Sarasota, FL) and then coated with a thin layer of acrylic. Animals were then immediately implanted with a vagus nerve stimulating cuff.

**Treatment group assignment and exclusion criteria**. Rats were dynamically allocated to balanced groups based on week 6 maximal pull forces to receive either rehabilitative training without VNS (Rehab), VNS paired with rehabilitative training (CL-VNS), VNS delivered at least 2 h after the last rehabilitative training session each day (Delayed VNS), or VNS paired with rehabilitative training following cortical cholinergic depletion (ACh-:CL-VNS). VNS was delivered only during weeks 7–11 and the 5-week CL-VNS pairings are similar to previous studies[24,25]. Experimenters were blind to the treatment group during testing and all behavioral analysis was automated to eliminate bias.

Forty-nine rats were excluded from this study based upon the following exclusion criteria: (1) Did not survive the peripheral nerve injury surgical procedure or VNS cuff implant ($N = 15$); (2) Did not display at least a 50% reduction in maximal pull force ($N = 6$); (3) Too impaired to freely perform the task following peripheral nerve injury ($N = 9$); (4) Headcap or stimulating cuff failure ($N = 19$). 30 of the 49 exclusions (1–3) were done prior to group assignment and thus could not impact the interpretation of results.

**Behavioral end measures**. At the conclusion of behavioral testing, additional behavioral assessments of motor and sensory function were performed. Testing was performed within 1 week of the conclusion of isometric pull task testing. All experimenters involved in behavioral testing and performing analysis were blinded to the treatment group of the animal.

**Forepaw mechanical sensory testing**. Somatosensory detection thresholds were assessed in a subset of animals (Rehab, $N = 12$; CL-VNS, $N = 12$; Delayed VNS, $N = 7$; ACh-:CL-VNS, $N = 4$) according to standard procedures[30]. Testing was performed in an acrylic chamber on a wire mesh floor. Animals were acclimated to the chamber the day before testing for 2 h. On the day of testing, measurements were taken following a 30 min acclimation period. Experimenters were blinded to the experimental group of the animal. Forepaw mechanical sensitivity was tested on the right and left forepaws using an Electronic Von Frey device (Ugo Basile, Dynamic Plantar Aesthesiometer). The actuator filament (0.5 mm diameter) was applied to the plantar surface of the forepaw, and a linearly increasing force was applied (20 s ramp time, 50 g maximal force). The force at which the paw withdrawal occurred was recorded for analysis. The left and right paw were alternately tested and with a minimum of 1 min interval between consecutive tests. Trials resulting in paw withdrawal due to spontaneous activity were excluded from analysis.

**Cylinder forelimb asymmetry test**. Spontaneous use of the forelimbs during exploratory activity was measured in a subset of animals (Rehab, $N = 12$; CL-VNS, $N = 12$; Delayed VNS, $N = 7$; ACh-:CL-VNS, $N = 5$). Animals were placed in a transparent cylinder and allowed to freely explore for two minutes. The cylindrical shape encouraged vertical rearing on to the wall of the cylinder. The video was recorded from directly underneath the cylinder through a clear sheet of acrylic. The total number of both left and right forepaw contacts with the wall of the cylinder were counted. The forepaw asymmetry index was then calculated [(contra/(contra + ipsi)) × 100].

**Intracortical microstimulation procedure**. Intracortical microstimulation mapping was used to document motor cortex movement representations after the conclusion of rehabilitative training in a subset of rats (Rehab, $N = 11$; CL-VNS, $N = 8$; Delayed VNS, $N = 7$; ACh-:CL-VNS, $N = 4$; Uninjured, $N = 4$). Rats were deeply anesthetized and a cisternal drain was performed to reduce ventricular pressure and cortical edema during mapping. A craniotomy was then performed to expose the left motor cortex. Low impedance tungsten microelectrodes (100 kOhm–1 MOhm electrode impedance; FHC Inc., Bowdin, MD) were inserted to a depth of 1.75 mm from the cortical surface and at an interpenetration resolution of 500 μm. Long-duration intracortical microstimulation pulses (ICMS) (biphasic ICMS at 333 Hz, 500 ms duration, 200 μs pulse duration, 0–200 μA current) were delivered in motor cortex[70]. Mapping experiments were performed with 2 experimenters both blinded to the treatment group of the animal. The first experimenter positioned the electrode for ICMS. The second experimenter, blind to both the experimental group of the animal and electrode position, delivered ICMS and classified movements. The stimulation current was increased from 1 μA until a movement was observed. If 200 μA was reached, the site was recorded as non-responsive. If a movement was elicited, the stimulating current to evoke the movement was recorded, then the ICMS current was increased by 50% (maximum 200 μA) to magnify the movement and facilitate visual classification[25,70]. Anecdotally, we observed that the movement elicited at threshold and 50% above threshold were the same; however, higher stimulation intensity evoked movements were more apparent and thus easier to accurately identify, consistent with previous reports[70]. All subjects underwent identical mapping procedures. No differences were observed across groups in the size of the motor map or the average stimulation intensity to evoke a movement (Supplementary Fig. 2). Movements were classified into the following categories: vibrissae, neck/jaw, digit flexion, digit extension, wrist extension, elbow flexion, shoulder, hindlimb, and trunk. The average cortical area (mm²) was calculated for each ICMS evoked movement for each group. Each electrode penetration site evoking a movement was counted as 0.25 mm². At electrode penetration sites evoking multiple movements, the movement area was divided by the number of evoked movements at that site.

**Pseudorabies virus injections and analysis**. Transsynaptic tracer injections were performed during Week 13 in a subset of rats (Rehab, $N = 5$; CL-VNS, $N = 5$). PRV-152 was a generous gift from the lab of Dr. Lynn Enquist and colleagues at Princeton University. Animals were deeply anesthetized and an incision was made over the medial face of the radius and ulna of the trained limb to expose the flexor digitorum profundus (FDP) and palmaris longus (PL). The FDP and PL are two main extrinsic forelimb grasping muscles, and both muscles receive joint innervation through the median and ulnar nerves[61]. 15 μL of PRV-152 was injected into the belly of each muscle in three separate injections of equal volume. The arm was then slightly rotated and the extensor carpi radialis longus (ECRL) was identified and exposed. The ECRL receives innervation solely through the radial nerve of the forelimb[61]. 20 μL of PRV-614 was injected in to the belly of the ECRL in four equal volume injections. The skin was then sutured with non-absorbable suture, rinsed with saline, and treated with antibiotic ointment. PRV-152 used in this study was ~1.5 × 10⁹ plaque-forming units (PFU), and the PRV-614 was ~9.05 × 10⁹ PFU. Rats were anesthetized with sodium pentobarbital (50 mg/kg, i.p.) and transcardially perfused with 4% paraformaldehyde in 0.1 M PBS (pH 7.5) at 144 h following injections. The brain was then removed, post-fixed overnight, then cryoprotected in 30% sucrose.

Brains were individually blocked and frozen at −80 °C in Shandon M1 embedding matrix (Thermo Fisher Scientific; Waltham, MA). Forebrain blocks were sectioned at 35 μm on a cryostat and immediately slide-mounted. Slides were coverslipped then scanned and digitized using the Virtual Slide Microscope VS120 (Olympus Corporation; Tokyo, Japan). Both PRV-152 (enhanced green fluorescent protein (EGFP) positive) and PRV-614 (monomeric red fluorescent protein (mRFP) positive) neuron counts were made on every other forebrain section (35 μm inter-slice interval). Sensorimotor cortical counts were restricted to layer V. All experimenters processing the tissue and cell counting were blinded to the treatment group of the animal.

**Electron microscopy evaluation of nerve morphology and analysis**. In a subset of animals (Rehab $N = 3$; CL-VNS $N = 3$; Delayed VNS $N = 3$) segments of the injured nerves proximal and distal to the injury site were removed for histo-pathological analysis. Under anesthesia, the proximal and distal segments of the median and ulnar nerves were identified. Animals were then perfused with 200 mL PBS followed by 200 mL 4% PFA. Segments (5–10 mm) were then dissected from both proximal and distal segments of the median and ulnar nerves. The nerve segments were then fixed in 4% PFA and 2.5% glutaraldehyde in 0.1 M cacodylate buffer for 2 h, then transferred to 2.5% glutaraldehyde in 0.1 M cacodylate buffer. Tissue was cut into 1 mm³ pieces and fixed with 2.5% (v/v) glutaraldehyde in 0.1 M sodium cacodylate buffer. Tissue samples were then rinsed in 0.1 M sodium cacodylate buffer and post-fixed in 1% osmium tetroxide and 0.8% potassium ferricyanide in 0.1 M sodium cacodylate buffer for 3 h at room temperature. After three rinses in water, the samples were then block stained with 4% uranyl acetate in 50% ethanol for 2 h. Samples were then dehydrated with increasing concentrations of ethanol, transitioned into resin with propylene oxide, infiltrated with Embed-812 resin and polymerized in a 60 °C oven overnight. Blocks were sectioned with a diamond knife (Diatome) on a Leica Ultracut 6 ultramicrotome (Leica Micro-systems) and collected onto copper grids, post stained with 2% aqueous Uranyl acetate and lead citrate. All experimenters handling and processing the tissue were blinded to the group of the animal.

Images were acquired on a Tecnai G² spirit transmission electron microscope (FEI) equipped with a LaB₆ source at 120 kV. Images were acquired at seven random non-overlapping locations throughout the nerve. All tissue preparation and imaging was performed at the UT Southwestern Electron Microscopy Core Facility (The University of Texas Southwestern Medical Center; Dallas, Texas). All analysis was performed using ImageJ. Analysis of number of myelinated fibers, axon area, fiber area, and g-ratio (axon area/fiber area) was performed at ×2550 magnification. The axon perimeter and myelin perimeter were traced and the areas were recorded. The number of axons completely enclosed within the image boundaries were then counted and recorded. Within one nerve segment, the average number of axons per unit area, average fiber diameter, and average g-ratio were calculated. G-ratio was calculated as the square root of axon area divided by fiber area. All imaging and analysis was performed by experimenters blinded to the experimental group.

**Quantification of ACh depletion**. In the subset of animals (ACh-:CL-VNS, $N = 5$) receiving 192-IgG-Saporin lesions and a subset of control animals ($N = 5$), animals were transcardially perfused and forebrains harvested as described above. The full extent of motor cortex was sectioned at 40 μm thickness. Three sections from motor cortex were randomly selected and stained for acetylcholinesterase (AChE) activity using a standard protocol[20]. Free floating sections underwent multiple washes in a Tris-Maleate buffer solution containing 6 mg/ml promethazine. Tissue was then incubated in a solution containing 10 mM sodium citrate, 30 mM cupric sulfate, 5.0 mM potassium ferricyanide, and 0.5 mg/ml acetylcholine iodide. Tissue was then washed in a Tris-HCl buffer and then processed for DAB to intensify labeling. Tissue was then slide mounted, dehydrated, and coverslipped.

Stained tissue was imaged on Virtual Slide Microscope VS120 (Olympus). Analysis of cortical cholinergic innervation was performed by counting AChE positive fibers crossing of a grid overlay[20]. A blinded experimenter randomly selected a region of layer V of motor cortex from each section, and a 6 × 6 grid (250 μm × 250 μm) was superimposed on the area. Intersections between AChE stained fibers and the gridline were identified and counted by a blinded experimenter. Depletion percentage (% Depleted; Supplementary Table 12) values were calculated as the number of fiber crossings per subject divided by the mean number of fiber crossings in controls subjects.

**Muscle fiber morphology analysis and quantification**. Following perfusion, the forelimb muscles (flexor digitorum profundus (FDP) and palmaris longus (PL)) were harvested, fixed in PFA, and cryoprotected in 30% sucrose for 24–48 h before embedding in Shandon M1 embedding matrix (Thermo Fisher Scientific; Waltham, MA). Transverse sections of 40 μm thickness were obtained and co-labeled with the NF200 antibody (1:200; Sigma Aldrich; St. Louis, MO) to visualize the heavy microfilament (200 kDa) in axons, and FITC-conjugated a-bungarotoxin (1:1000; Invitrogen; Carlsbad, CA) to visualize the nicotinic acetylcholine receptors (AChR). The images are presented with maximum intensity projection and the muscle fiber area was quantified by measuring the area of individual fibers using ImageJ.

**Statistics**. All comparisons were planned in the experimental design a priori. Normality of distributions were tested using Lilliefors tests (Supplementary Table 20), and significant differences for parametric comparisons were determined using one-way ANOVAs followed by two-sided unpaired $t$-tests (Fig. 2a, Digit Flexion; Figs. 3d; 4b; 5e), and two-way repeated measures ANOVAs followed by unpaired $t$-tests (Figs. 3a, b; 5c). Significant differences for nonparametric comparisons were determined using Kruskal–Wallis tests followed by Wilcoxon rank-sum tests (Fig. 2a, Extension; Figs. 2c; 4d, g; 5a). Alpha of 0.05 was used for single comparisons (Figs. 4e–g; 5a). To correct for multiple comparisons, a Bonferroni-corrected alpha of 0.025 was used (Figs. 2a, c; 3a, b, d; 4b; 5c, e). Statistical tests for each comparison are noted in the text and all statistical tests for all figures are reported in Supplementary Tables 18–20. Statistical analysis was performed in MATLAB 2016b. Unless otherwise noted, * indicates $p < 0.025$, ** indicates $p < 0.005$, and *** indicates $p < 0.0005$. Error bars indicate mean ± SEM in all figures.

**Reporting summary**. Further information on research design is available in the Nature Research Reporting Summary linked to this article.

## Data availability
All data generated during and/or analyzed during the current study are available in the supplementary materials. Analysis code is available upon request. Electron microscopy images are available at the following links: https://doi.org/10.6084/m9.figshare.8239751, https://doi.org/10.6084/m9.figshare.8239694, https://doi.org/10.6084/m9.figshare.8239754. The source data underlying Figs. 2a–d, 3a–d, 4b, d, e–g, 5a–e, Supplementary Figs. 3, 5, 7, 9, 10, 11, 14, 15, 17, 18, and 20 are provided as a Source Data file.

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

## Acknowledgements

The authors thank David Pruitt for engineering assistance, Kimiya Rahebi for electronics construction, and Katherine Adcock, Nicole Robertson, Sharanya Ajaykumar, and Justin James for assistance with mapping procedures, and Daniel Hulsey, Andrea Ruiz, Han-saim Jeong, and Tracey Phan for help with histological procedures. In addition, they would like to thank Preston D'Souza, Sanketh Kichena, Sania Khan, Sina Kashef, for help with behavioral testing. This work was supported by NIH R01 NS085167 and R01 NS094384, and by the Defense Advanced Research Projects Agency (DARPA) Biological Technologies Office (BTO) Electrical Prescriptions (ElectRx) program under the auspices of Dr. Eric Van Gieson through the Space and Naval Warfare Systems Center, Pacific Cooperative Agreement No. N66001-15-2-4057 and the DARPA BTO Targeted Neu-roplasticity Training (TNT) program under the auspices of Dr. Tristan McClure-Begley through the Space and Naval Warfare Systems Center, Pacific Grant/Contract No. N66001-17-2-4011.

## Author contributions

E.C.M., N.K., B.R.S., E.L., A.B., P.G., and G.B. performed experiments. E.C.M and S.A.H. analyzed data. E.C.M., M.R.O., R.L.R., M.P.K., and S.A.H. designed the experiments and wrote the paper.

## Competing interests

M.P.K. has a financial interest in MicroTransponder, Inc., which is developing VNS for stroke and tinnitus. R.L.R. is a co-owner of Vulintus, Inc., which makes rodent behavioral testing systems, and owner of Teliatry, which is developing a VNS device. All other authors declare no conflict of interest.

**Additional information**

