## [Peer Review File · Nature Communications]

Reviewers' comments:

Reviewer #1 (Remarks to the Author):

This paper reports the results of a study to examine the effect of vagal nerve stimulation on cortical map plasticity and recovery of motor function in rats, subsequent to peripheral nerve injury. The study uses an impressive combination of approaches to relate the specific effects of nerve damage and the role of cholinergic systems in promoting recovery to both typical cortical maps and restored motor behavior. The general findings are of great interest with both basic and clinical impact. They stem from a long line of research linking cortical map architecture, cholinergic systems, and behavioral motor recovery, but this paper ties together the three in a novel way. Anatomical measures, behavior and electrical stimulation mapping to attempt to link abnormal changes in the map to behavioral recovery and to reveal the underpinning role of the cholinergic in this effect. However, there are a number of issues connected with the way the results are reported that require attention because the claims exceed the results. In addition, the figures are very dense and it is difficult to evaluate all of the data based on the display style and compactness. Finally, the statistical methods used do not seem to be appropriate.

Here, the result of peripheral nerve injury is persistent abnormal functional architecture of cortical motor maps as well as ongoing behavioral deficits in arm movement that may be repaired by VNS. The main findings show that the change in the size of cortical representations of microstimulation evoked movements can be modified by VNS, in a way that restores the control pattern in motor cortex and that this correlates with behavioral recovery on a reach and grasp task. This is an interesting and noteworthy result. The study also shows that this effect is blocked by a cholinergic toxin and that changes in the density of corticospinal connections is consistent with a cortical basis for this effect. Finally, the investigators indicate that muscles and regenerated nerves are unlikely to account for behavioral effect, so that motor behavioral deficits may be the consequence of cortical changes. The paper also shows that blocking ACh eliminates the VNS effect on flexors, but appears to accentuate it in extensors, which the authors call "pathological plasticity" that may be a mechanism by which this cortical plasticity is regulated. The authors generally make statements that are too strong for the data, but they are nevertheless a clever mix of methods that help cement the relationships between cortical architecture and behavior, injury and cholinergic systems that can be controlled by cholinergic systems.

Specific comments on the report:

Nerve injury: Given that nerves regenerate about 1mm/day, it would be expected that the reinnervation would be just beginning to reach the denervated stump at the time measurement are first made and that mapping is being done during the time when afferents and efferents in the periphery are still reinnervating. What is the state of the system? Is this reinnervation speed correct for rats? What is the expected behavioral timecourse of this effect and how would this affect the behavioral measures as well as conclusions about central vs. peripheral mechanisms?

I am puzzled why the muscles are called grasp and extensors. Would it be better to call 'grasp' flexion to match 'extension'. Both involve forearm and finger muscles; one term is a behavior the other a muscle action.

The results regarding map changes should be reported as increasing the area of motor cortex where grasp could be evoked (the experiment does not demonstrate that this is the area "controlling ...grasp network". Likewise, elimination of the maladaptive...." Is also an overstatement and needs to be tempered and made more objective. Fig. 1g does not demonstrate the emergence of abnormal muscle synergies, but a change in the movements evoked by highly non-physiological electrical stimulation of the motor cortex. EMG recordings of muscle synergies were not measured

and thus this conclusion should be properly restated.

P. 4 why at the end of the 1st paragraph on this page does the involvement of ACh system follow from the logic of the prior statements?

P. 3. Bursts of VNS stimulation is vague. It would be useful for the reader here to have some explanation of the stimulation pattern and duration. What is the rationale for this pattern? Ideally (but impractical now) the effect of other patterns would be useful to know and should not be a reason to reject this study).

Fig 3 mentions synaptic reorganization, when the measure is # of neurons, which they could interpret to mean reorganization, but this is not demonstrated. They further did not demonstrate here 'connectivity' but again # of neurons. This can be related to injection size, effectiveness of uptake mechanisms, etc. The authors need more cautious verbiage here. In d, what is the time point of the distal photo. In F, what is the stain? The point of this figure should be made.

This data in this figure and others reveal an issue with the statistical approaches used when you have such a small sample and a non-normal distribution of the data. The bar graphs in e and g have so little data that representing them as a bar with means and variance is problematic; it is helpful that individual data points are shown and a better way to display such data should be considered. Further, appropriate statistical methods should be used as for this type of data. They will likely show the effect, I suspect, but the outliers and distribution should be addressed (they are lost in the use of these descriptive displays).

Pathological plasticity is another interpretation that should be changed to say what the measure is, and what it shows, not the authors' interpretation. All of these interpretations should be saved for the final figure and a discussion outside of the results reporting.

Fig 4 a should be in the same layout as fig 1. That is the right bar graph shows a repeat of 1e but flipped and with ACh effect (the first point of the figure) This would make it easier to follow. The reader should be directed to compare 4 with 1.

Cholinergic depletion: experiments combine mapping and cholinergic depletion with a specific toxin. However, like all such toxins, the effects are not necessarily precise and other systems may be affected. This is not a fatal flaw, but it is worthy of commentary.

Dement Geriatr Cogn Disord. 2011;32(1):70-8. doi: 10.1159/000330741. Epub 2011 Aug 26.

Decrease of GABAergic markers and arc protein expression in the frontal cortex by intraventricular 192 IgG-saporin. Jeong DU1, Chang WS, Hwang YS, Lee D, Chang JW.

Fig. 5 should read postulated/proposed mechanism (supported by these data).

The explanations are plausible, but of course there are many caveats: e.g. the vagus stimulation will activate many CNS pathways (and release hormones in its end organs, change heart rate, etc) that might also contribute to this effect. Most have not been ruled out by these experiments.

P. 5 The authors link number of neurons to drive from motor cortex (1e). This is not necessarily true since drive can be increased without more neurons (e.g. higher firing rates).

The number of neurons is called synaptic connectivity, but it is not necessarily so. The measure should be stated as it is: # neurons. The impact of that on recovery would then be appropriate.

In sensory threshold measures, the use of means and SEM here as well as elsewhere is somewhat questionable. Note in 2F that three of the Delayed VNS were as recovered as 3 of the CL VNS animals. Is there anything noteworthy about the best recovering animals?

It would also be useful to present the distribution of results for success rate and pull force. While

the means give a nice clear illustration of the effect, perhaps a description of how these means fit the actual data would be informative (i.e. do they all show this trend or is there variability, how much?). Pull force seems to be even more concerning in this realm. Was there a super performer in the group? This criticism could be overcome by a description in the text about the 'normalcy' of the effect and some measures of it.

Reviewer #2 (Remarks to the Author):

This is a very comprehensive and well-designed study demonstrating that vagus nerve stimulation that is timed with rehabilitative approach induces plasticity associated with recovery. The authors cleverly used multimodal techniques ranging from the a battery of animal behavior to electrophysiology, microscopy and cellular measurements. This builds on prior evidence showing that the peripheral nerve injury as well as other type of injuries can often lead to maladaptive plasticity that can delay rehabilitation. Of particular interest is the exact timing of the vagus stimulation, that could lead to a shift in clinical strategies.

Overall I am very impressed with this work and would suggest only a couple of minor revisions:

1. It was not clear if the vagus nerve stimulation provides unilateral or bilateral stimulation, and where was the calf stimulate placed in relation to the deafferented cortex?
2. The acetylcholine hypothesis is interesting, but perhaps would be interesting to discuss other neurotransmitters playing into that post-injury plasticity. For example, there are previous works demonstrating increased in GABA after peripheral injury.
3. There is growing evidence suggesting that brain stimulation paired with other means of rehabilitation facilitates recovery, not only in peripheral nerve injury but in animal models of traumatic brain injury (PMID: 30082198) and spinal cord injury (Front. Neurosci. | doi: 10.3389/fnins.2019.00387). These references also support the timing specificity phenomenon that the current paper suggests.

Reviewer #3 (Remarks to the Author):

Meyers et al., investigate the use of closed-loop vagus nerve stimulation (CL-VNS) in addition to rehabilitation to restore sensory and motor function in the forelimb. Previous studies have implemented CL-VNS to enhance central plasticity for stroke and spinal cord injury. This study tests these findings to influence peripheral nerve transection. From this study, they show that they are able to achieve reversal of maladaptive central plasticity caused by severe nerve damage to recover sensory and motor function without peripheral changes. The performance of the CL-VNS was assessed through, recovery of forelimb function, anatomical, and physiological studies. The manuscript details important findings for vagus nerve stimulation to influence recovery from peripheral nerve injury through central plasticity; however, there are some concerns with the manuscript in its current form. Below are some items that need to be addressed:

Major Comments:

- 1) Fig. 1 (f): Why was data not collected for uninjured animals? Also Fig. 1 (h).
- 2) Fig. 2 (c): What is the timeline for recovery in this image? It appears that they all regain 100% recovery by the end.
- 3) Page 6, second paragraph (line 143). Although there are no changes histologically in nerves with VNS vs. nerves without VNS, this does not give the ability to fully conclude that there are no peripheral changes. Peripheral electrophysiological data would help to confirm whether this is the case.
- 4) Fig. 3 (d) It is mentioned that the distal sections show some reinnervation but still have impaired architecture. This is not very clear in these images. There is no explanation for how the peripheral nerve architecture is affected by CL-VNS (both the tissue from Rehab and CL-VNS

groups looks the same).

5) It would be helpful to have histology post-mortem of each group of rats to assess regeneration and influence of each treatment.

6) Was the same amount of current delivered to the delayed VNS group? If so, it is important to explicitly state this.

7) Was functional recording data for the cuffs collected? How was assurance of current delivery achieved?

8) It would be helpful to determine functionality and overall assessment of nerve cuffs post-mortem also through images of the devices post-mortem.

9) Fig. S4 seems more significant by including uninjured animal data that what is currently provided in Fig. 2(a,b)

10) The manuscript would benefit from showing more histological data. Tissue from different end points in the study showing axons, macrophages, cell nuclei, and potentially staining for different types of fibers or myelinated vs unmyelinated fibers would help elucidate the anatomical aspects of the central changes.

Minor Comments:

1) Page 4-5: Citations are missing for the following statement: "these results are consistent with classical studies documenting the reduction in injured circuits and a consequent expansion of spared circuits..."

2) Page 5: Delete "if" in "Here, we tested the if CL-VNS paired with reach-and-grasp training..."

3) Line 145: it is important here to indicate that the strength impairment is in the Rehab and delayed VNS groups only.

4) The way the sub figures are labeled is not consistent in all figures.

5) Fig. 5 is not discussed adequately in the manuscript

6) Methods: why was the study conducted only in female rats?

Dear Editors and Reviewers,

We thank you for the opportunity to revise our manuscript entitled “Enhancing plasticity in central networks improves motor and sensory recovery after nerve damage” (NCOMMS-19-07045). We have made changes based on the reviewers’ comments, which has significantly improved the manuscript. Below we address the reviewers’ concerns item by item. Given the high clinical incidence of peripheral neuropathy and growing interest in non-pharmacological interventions, we believe that this paper will advance thinking in the field and will be highly cited for years to come.

Blue text in the revised main manuscript indicates changes.

Reviewer #1

- 1. Nerve injury: Given that nerves regenerate about 1mm/day, it would be expected that the reinnervation would be just beginning to reach the denervated stump at the time measurement are first made and that mapping is being done during the time when afferents and efferents in the periphery are still reinnervating. What is the state of the system? Is this reinnervation speed correct for rats? What is the expected behavioral timecourse of this effect and how would this affect the behavioral measures as well as conclusions about central vs. peripheral mechanisms?***

Nerve regeneration in the rat is reported between 1-3.5 mm/day (1). Based on these regeneration speeds and the location of the injury, we expected to see return of function to the affected limb at around 4 weeks post-injury. In earlier pilot studies (2), we confirmed this and observed that animals were capable of freely performing the isometric pull task beginning at week 4 post-injury. While this is certainly the initial phase of regeneration, we chose to include an additional 2 weeks and begin rehabilitation at 6 weeks post-injury in an attempt to match the clinical standard of initiating rehabilitation at the first signs of reinnervation (3).

The peripheral nervous system is still in a state of regeneration and reinnervation throughout the entire course of the study, as reinnervation and remyelination continues for many months following nerve transection and repair (4). However, we observed asymptotic motor performance around week 10-11 in the rehabilitation alone group, suggesting that although reinnervation/remyelination is still occurring, the majority of regenerated axons have reinnervated end-targets. This corroborates previous studies using a similar injury in which subjects reached asymptotic motor performance at 6 weeks post-injury (5, 6) (although subjects did not undergo intensive rehabilitation such as in the present study). Compared to this previous study, the recovery observed in Rehab subjects here is likely both a function of intensive rehabilitation and continued axonal regeneration. The central nervous system is rapidly adapting to continued regeneration (7), and since both rehabilitation and VNS is delivered during this period of regeneration, it is likely that we are augmenting these central processes. We have now included additional discussion of this topic in the manuscript (pg. 4).

- 2. I am puzzled why the muscles are called grasp and extensors. Would it be better to call ‘grasp’ flexion to match ‘extension’. Both involve forearm and finger muscles; one term is a behavior the other a muscle action.***

We agree with the reviewer and have revised the term “grasp” to digit flexion to match the extensor muscle action concept.

- 3. The results regarding map changes should be reported as increasing the area of motor cortex where grasp could be evoked (the experiment does not demonstrate that this is the area “controlling ...grasp network”. Likewise, elimination of the maladaptive....” Is also an overstatement and needs to be tempered and made more objective. Fig. 1g does not demonstrate the emergence of abnormal muscle synergies, but a change in the movements evoked by highly non-physiological electrical stimulation of the motor cortex. EMG recordings of muscle synergies were not measured and thus this conclusion should be properly restated.***

We agree with the reviewer and have modified the relevant statements accordingly (pg. 5-6).

- 4. P. 4 why at the end of the 1st paragraph on this page does the involvement of ACh system follow from the logic of the prior statements?***

We thank the reviewer for identifying that oversight in the previous version of the manuscript. We have now removed that section from the updated manuscript (pg. 5).

- 5. P. 3. Bursts of VNS stimulation is vague. It would be useful for the reader here to have some explanation of the stimulation pattern and duration. What is the rationale for this pattern? Ideally (but impractical now) the effect of other patterns would be useful to know and should not be a reason to reject this study).***

We agree with the reviewer that the previous version of the manuscript lacked adequate discussion of VNS stimulation delivery and parameters. To remedy this, we have now included a detailed description of VNS parameters and evidence to justify the selection of the parameters utilized in this study (pg. 4, 17).

- 6. Fig 3 mentions synaptic reorganization, when the measure is # of neurons, which they could interpret to mean reorganization, but this is not demonstrated. They further did not demonstrate here ‘connectivity’ but again # of neurons. This can be related to injection size, effectiveness of uptake mechanisms, etc. The authors need more cautious verbiage here.
In d, what is the time point of the distal photo. In F, what is the stain? The point of this figure should be made.***

We agree with the reviewer and have made the Figure 3 (now Figure 4) legend description more objective. Furthermore, we have modified the Results section (pg. 8) and included additional discussion about the interpretation and limitations of the tracing results described in Figure 3 (pg. 15-16).

We apologize for the mistake in Fig. 3D and F and have corrected it in the updated manuscript figure legend. The distal photo in Fig. 3D was taken at Week 13 after injury (end of the study), and muscle fibers in Fig. 3F were visualized using non-specific background fluorescence.

- 7. This data in this figure and others reveal an issue with the statistical approaches used when you have such a small sample and a non-normal distribution of the data. The bar graphs in e and g have so little data that representing them as a bar with means and variance is problematic; it is helpful that individual data points are shown and a better way to display such data should be considered. Further, appropriate statistical methods should be used as for this type of data. They will likely show the effect, I suspect, but the outliers and distribution should be addressed (they are lost in the use of these descriptive displays).**

We appreciate the feedback on these figures from the reviewer. We have updated the statistical tests used in Fig. 4e to nonparametric tests to account for the small sample sizes. Additionally, we have now included discussion in the limitations section of the manuscript to discuss the high variance in the EM data that could potentially explain the behavioral effects (pg. 17). All of the data from individual subjects from the electron microscopy studies are now included in the supplementary materials, along with links (Acknowledgments, pg. 26) to the data repository that hosts all EM images used for analysis (Supplementary Materials Table S5, 6, 16, 17).

- 8. Pathological plasticity is another interpretation that should be changed to say what the measure is, and what it shows, not the authors' interpretation. All of these interpretations should be saved for the final figure and a discussion outside of the results reporting.**

We have removed all references to the pathological plasticity interpretation from the Results section. Furthermore, we have included additional discussion on the interpretation of these results in the Discussion section (pg. 12).

- 9. Fig 4 a should be in the same layout as fig 1. That is the right bar graph shows a repeat of 1e but flipped and with ACH effect (the first point of the figure) This would make it easier to follow. The reader should be directed to compare 4 with 1.**

We agree that the inconsistency between these two figures might potentially lead to unnecessary confusion. We have modified Figure 4a/b (now Figure 5a/b) so that it matches the layout of Figure 1e (now Figure 2a/b) and have directed readers to compare to Figure 2.

- 10. Cholinergic depletion: experiments combine mapping and cholinergic depletion with a specific toxin. However, like all such toxins, the effects are not necessarily precise and other systems may be affected. This is not a fatal flaw, but it is worthy of commentary. Dement Geriatr Cogn Disord. 2011;32(1):70-8. doi: 10.1159/000330741. Epub 2011 Aug 26. Decrease of GABAergic markers and arc protein expression in the frontal cortex by intraventricular 192 IgG-saporin. Jeong DU1, Chang WS, Hwang YS, Lee D, Chang JW.**

We agree with the reviewer that toxin specificity is an important consideration that the previous version of the manuscript did not address. Additional discussion of this topic is now included in the Discussion section of the manuscript (pg. 17).

- 11. Fig. 5 should read postulated/proposed mechanism (supported by these data). The explanations are plausible, but of course there are many caveats: e.g. the vagus**

stimulation will activate many CNS pathways (and release hormones in its end organs, change heart rate, etc) that might also contribute to this effect. Most have not been ruled out by these experiments.

We agree that there are other explanations that cannot be ruled out, and we have modified the title of Figure 6. Furthermore, we have modified the discussion to discuss other possible mechanisms that might contribute to CL-VNS enhancement of recovery (pg. 17).

12. P. 5 The authors link number of neurons to drive from motor cortex (1e). This is not necessarily true since drive can be increased without more neurons (e.g. higher firing rates).

The number of neurons is called synaptic connectivity, but it is not necessarily so. The measure should be stated as it is: # neurons. The impact of that on recovery would then be appropriate.

We agree with the reviewer and have modified those statements to specifically state number of cortical neurons to avoid misinterpretation (pg. 8).

13. In sensory threshold measures, the use of means and SEM here as well as elsewhere is somewhat questionable. Note in 2F that three of the Delayed VNS were as recovered as 3 of the CL VNS animals. Is there anything noteworthy about the best recovering animals? It would also be useful to present the distribution of results for success rate and pull force. While the means give a nice clear illustration of the effect, perhaps a description of how these means fit the actual data would be informative (i.e. do they all show this trend or is there variability, how much?). Pull force seems to be even more concerning in this realm. Was there a super performer in the group? This criticism could be overcome by a description in the text about the 'normalcy' of the effect and some measures of it.

We appreciate the feedback from the reviewer on the representation of this dataset. We now report the normality of the distributions using Lilliefors tests along with all non-parametric and parametric statistical tests in the supplement (Supplementary Table 20). The Statistics section has now been revised and directs readers to those Supplementary Tables. Furthermore, we now use non-parametric statistical tests where appropriate (Supplementary Materials pg. 14-15).

Anecdotally, we did not observe any noteworthy features in animals that demonstrate greater recovery. Additionally, correlation analysis of pull force and sensory thresholds or ICMS data fail to indicate any significant association between parameters, which are now included in Supplementary Table 21. We report all of the individual data and statistical tests in the supplementary tables to allow readers to explore additional associations.

Reviewer #2

- 1. It was not clear if the vagus nerve stimulation provides unilateral or bilateral stimulation, and where was the calf stimulate placed in relation to the deafferented cortex?***

We thank the reviewer for this important point which was not addressed in the previous version of the manuscript. Cuff electrodes were placed on the left cervical vagus nerve and the right forelimb was deafferented, thus predominantly affecting the left cortex. Vagal fibers synapse bilaterally on neurons within the nucleus tractus solitarius that then project bilaterally to the cholinergic basal forebrain and noradrenergic locus coeruleus (8–12). While in the present study the cuff electrode as placed on the same side as the deafferented cortex, previous studies from our lab have documented that CL-VNS improves recovery in bilateral injury models such as bilateral spinal cord injury and tinnitus (13, 14), consistent with the bilateral actions of VNS. Furthermore, recent clinical data in stroke survivors demonstrates that left cervical VNS is effective in improving motor function in patients with infarcts in either the right or left hemisphere (15). We have now added discussion on this topic in the Discussion section of the manuscript (pg. 12-13).

- 2. The acetylcholine hypothesis is interesting, but perhaps would be interesting to discuss other neurotransmitters playing into that post-injury plasticity. For example, there are previous works demonstrating increased in GABA after peripheral injury.***

We agree with the reviewer that additional discussion of neurotransmitters other than acetylcholine and their role in post-injury plasticity would greatly benefit the manuscript. We have now included additional discussion on this topic (pg. 14).

- 3. There is growing evidence suggesting that brain stimulation paired with other means of rehabilitation facilitates recovery, not only in peripheral nerve injury but in animal models of traumatic brain injury (PMID:30082198) and spinal cord injury (Front. Neurosci. | doi: 10.3389/fnins.2019.00387). These references also support the timing specificity phenomenon that the current paper suggests.***

We appreciate the feedback on this critical feature of neuromodulation therapeutics and have now included additional discussion on timing specificity of brain and spinal cord stimulation for improving functional outcomes (pg. 12-13).

Reviewer #3

1. Fig. 1 (f): Why was data not collected for uninjured animals? Also Fig. 1 (h).

We collected data in uninjured animals, but it was omitted from the initial submission to simplify the presentation of the figures. We have now added bubble plots including the uninjured subjects as a supplementary figure (Supplementary Figure 2).

2. Fig. 2 (c): What is the timeline for recovery in this image? It appears that they all regain 100% recovery by the end.

We apologize for the confusion surrounding Fig. 2c (now Fig. 3c). We have now modified the figure to clarify that the shaded green blocks denote differing percentages of recovery for each subject at Week 12. We have also modified the figure caption to clarify how the percent recovery value was calculated.

3. Page 6, second paragraph (line 143). Although there are no changes histologically in nerves with VNS vs. nerves without VNS, this does not give the ability to fully conclude that there are no peripheral changes. Peripheral electrophysiological data would help to confirm whether this is the case.

We agree with the reviewer that based on the data collected in this study, we are unable to fully conclude that VNS does not result in peripheral changes. As mentioned, one limitation of the current study is the lack of peripheral electrophysiological data that corroborates the anatomical findings and demonstrates nerve and muscle physiology is unchanged in response to VNS treatment. We have recently collected electrophysiological data in a different pilot study using a similar injury model that demonstrates that onset latency of neural activity in the somatosensory cortex in response to tactile stimulation of the paw is increased after nerve injury but is unchanged after CL-VNS treatment (Figure below). These preliminary results suggest that VNS does not alter nerve conduction velocity and are consistent with the absence of VNS-dependent peripheral changes. To address these concerns of the reviewer, we have added further discussion of these limitations in the Discussion section of the manuscript (pg. 17).

- 4. Fig. 3 (d) It is mentioned that the distal sections show some reinnervation but still have impaired architecture. This is not very clear in these images. There is no explanation for how the peripheral nerve architecture is affected by CL-VNS (both the tissue from Rehab and CL-VNS groups looks the same).**

We apologize for the confusion surrounding the influence of CL-VNS on peripheral nerve architecture. We did not observe changes of nerve architecture in CL-VNS subjects compared to both Rehab or Delayed VNS subjects. To improve the clarity of this figure for readers, we have rewritten the figure legend to emphasize that no statistical differences were observed between the VNS and Rehab groups. Additionally, we include a phrase to point the reader to the supplemental information for more detail on nerve histology (pg. 7).

- 5. It would be helpful to have histology post-mortem of each group of rats to assess regeneration and influence of each treatment.**

We agree with the reviewer that a clear description of the histology is necessary to interpret the results of the study. Post-mortem histology was performed on subjects in each group at Week 13 post-injury, within a week following the conclusion of behavioral testing. Analysis of nerve histology using electron microscopy was performed on tissue from subjects in the CL-VNS, Rehab, and Delayed VNS groups at this time-point (Week 13). We now include additional description in the manuscript of when the histology was performed (pg. 7) and include a link to a data repository containing all of the images used for analysis in the study (Delayed VNS: <https://doi.org/10.6084/m9.figshare.8239754>; CL-VNS: <https://doi.org/10.6084/m9.figshare.8239751>; Rehab: <https://doi.org/10.6084/m9.figshare.8239694>)

- 6. Was the same amount of current delivered to the delayed VNS group? If so, it is important to explicitly state this.**

Yes, all stimulation parameters were matched in the CL-VNS and Delayed VNS groups, only the timing of VNS delivery was varied. We apologize for this oversight in the previous version of the manuscript and have now provided a detailed description of the VNS parameters for all groups in the main manuscript (pg. 4).

- 7. Was functional recording data for the cuffs collected? How was assurance of current delivery achieved?**

Cuff impedance was monitored daily and animals were removed if cuff impedance exceeded 15 k Ω , consistent with previous studies, to ensure current delivery was effectively achieved throughout therapy. Furthermore, to evaluate activation of the vagus nerve at the conclusion of the study, we assessed activation of the Hering-Breuer (HB) reflex, a well-established biomarker of A-fiber activation in the vagus (16). To do so, 5-second trains of VNS were delivered under anesthesia and stimulation-dependent reduction of blood oxygen saturation was measured. We now include greater detail on assessment of cuff functionality in the manuscript (pg. 4).

- 8. It would be helpful to determine functionality and overall assessment of nerve cuffs post-mortem also through images of the devices post-mortem.**

While visual assessment of the stimulation cuffs provides a means to identify some failure modes, we elected to use monitoring of cuff impedance and activation of the Hering-Breuer reflex as described above. These measures represent the most widely used and well-validated markers and provide a thorough characterization of VNS cuff functionality in all animals throughout the study.

We now report the cuff impedance for all subjects that received VNS in Supplementary Table 22 (Means \pm SEM: CL-VNS: 5.78 ± 0.42 K Ω ; Delayed VNS: 5.47 ± 0.54 K Ω ; ACh::CL-VNS: 5.14 ± 0.89 K Ω).

9. *Fig. S4 seems more significant by including uninjured animal data that what is currently provided in Fig. 2(a,b)*

We appreciate the feedback on Figure 2a,b and have now improved the figure by including the uninjured animal data (now Fig. 3a/b).

10. *The manuscript would benefit from showing more histological data. Tissue from different end points in the study showing axons, macrophages, cell nuclei, and potentially staining for different types of fibers or myelinated vs unmyelinated fibers would help elucidate the anatomical aspects of the central changes.*

We agree with the reviewer that additional histological data would elucidate many of the anatomical aspects underlying the central changes that contribute to recovery. Although multiple anatomical metrics were documented in this study (nerve morphology, viral tracing, muscle fiber morphology), further anatomical investigation studies merit consideration and are central to our ongoing follow-up work. As such, we now provide links in the manuscript to all of the electron microscopy images of nerve tissue proximal and distal in VNS and no VNS rats, and include discussion on this topic in the Discussion section detailing the limitations of the study and future directions of further research (pg. 17).

Minor Comments

11. *Page 4-5: Citations are missing for the following statement: “these results are consistent with classical studies documenting the reduction in injured circuits and a consequent expansion of spared circuits...”*

We thank the reviewer for identifying this mistake and we have corrected it (pg. 5).

12. *Page 5: Delete “if” in “Here, we tested the if CL-VNS paired with reach-and-grasp training...”*

This mistake has been corrected (pg. 5).

13. *Line 145: it is important here to indicate that the strength impairment is in the Rehab and delayed VNS groups only.*

We have reworded that sentence to clarify the strength impairment in Rehab subjects.

14. *The way the sub figures are labeled is not consistent in all figures.*

We have fixed the figure panel labeling, and it is now consistent across figures.

15. *Fig. 5 is not discussed adequately in the manuscript*

We have now included additional references to Figure 5 (now figure 6) in the Discussion section of the manuscript.

16. *Methods: why was the study conducted only in female rats?*

Female rats were used due to ease of handling, and because the behavioral measures (17, 18), lesion model (2), and VNS parameters (13, 19–23) have all been extensively optimized in this sex. The Methods section has been modified to include this information.

References

1. T. M. Brushart *et al.*, Electrical stimulation promotes motoneuron regeneration without increasing its speed or conditioning the neuron. *J. Neurosci.* **22**, 6631–6638 (2002).
2. E. C. Meyers *et al.*, Median and ulnar nerve injuries reduce volitional forelimb strength in rats. *Muscle and Nerve*. **56** (2017), doi:10.1002/mus.25590.
3. M. D. Robinson, S. Shannon, Rehabilitation of peripheral nerve injuries. *Phys. Med. Rehabil. Clin. N. Am.* **13**, 109–135 (2002).
4. G. Melli, A. Höke, Canadian Association of Neurosciences Review: Regulation of Myelination by Trophic Factors and Neuron-Glial Signaling. *Can. J. Neurol. Sci. / J. Can. des Sci. Neurol.* **34**, 288–295 (2007).
5. C. M. Galtrey, R. A. Asher, F. Nothias, J. W. Fawcett, Promoting plasticity in the spinal cord with chondroitinase improves functional recovery after peripheral nerve repair. *Brain*. **130**, 926–39 (2007).
6. C. M. Galtrey, J. W. Fawcett, Characterization of tests of functional recovery after median and ulnar nerve injury and repair in the rat forelimb. *J. Peripher. Nerv. Syst.* **12**, 11–27 (2007).
7. X. Navarro, M. Vivó, A. Valero-Cabré, Neural plasticity after peripheral nerve injury and regeneration. *Prog. Neurobiol.* **82**, 163–201 (2007).
8. M. S. George *et al.*, Vagus nerve stimulation: a new tool for brain research and therapy. *Biol. Psychiatry*. **47**, 287–95 (2000).
9. G. G. Berntson, M. Sarter, J. T. Cacioppo, Anxiety and cardiovascular reactivity: the basal forebrain cholinergic link. *Behav. Brain Res.* **94**, 225–248 (1998).
10. T. R. Henry, Therapeutic mechanisms of vagus nerve stimulation. *Neurology*. **59**, S3–S14 (2002).
11. K. Semba, P. B. Reiner, E. G. McGeer, H. C. Fibiger, Brainstem afferents to the magnocellular basal forebrain studied by axonal transport, immunohistochemistry, and electrophysiology in the rat. *J Comp Neurol.* **267**, 433–453 (1988).
12. E. J. Van Bockstaele, J. Peoples, P. Telegan, Efferent projections of the nucleus of the solitary tract to peri-Locus coeruleus dendrites in rat brain: Evidence for a monosynaptic pathway. *J. Comp. Neurol.* **412**, 410–428 (1999).
13. P. D. Ganzer *et al.*, Closed-loop neuromodulation restores network connectivity and motor control after spinal cord injury. *Elife*. **7**, 1–19 (2018).
14. N. D. Engineer *et al.*, Reversing pathological neural activity using targeted plasticity. *Nature*. **470**, 101–4 (2011).
15. J. Dawson *et al.*, Safety, Feasibility, and Efficacy of Vagus Nerve Stimulation Paired With Upper-Limb Rehabilitation After Ischemic Stroke. *Stroke*. **47**, 143–50 (2016).
16. M. Rios *et al.*, Protocol for Construction of Rat Nerve Stimulation Cuff Electrodes. *Methods Protoc.* **2**, 19 (2019).
17. S. A. Hays *et al.*, The isometric pull task: a novel automated method for quantifying forelimb force generation in rats. *J. Neurosci. Methods*. **212**, 329–37 (2013).
18. A. M. Sloan *et al.*, A Within-Animal Comparison of Skilled Forelimb Assessments in Rats. *PLoS One*. **10**, e0141254 (2015).
19. K. W. Loerwald, M. S. Borland, R. L. Rennaker, S. A. Hays, M. P. Kilgard, The interaction of pulse width and current intensity on the extent of cortical plasticity evoked by vagus nerve stimulation. *Brain Stimul.* **11**, 271–277 (2018).
20. K. W. Loerwald *et al.*, Varying Stimulation Parameters to Improve Cortical Plasticity Generated by VNS-tone Pairing. *Neuroscience*. **388**, 239–247 (2018).
21. E. P. Buell *et al.*, Cortical map plasticity as a function of vagus nerve stimulation rate. *Brain Stimul.* **11**, 1218–1224 (2018).
22. D. R. Hulseley *et al.*, Parametric characterization of neural activity in the locus coeruleus in response to vagus nerve stimulation. *Exp. Neurol.* **289**, 21–30 (2017).
23. M. S. Borland *et al.*, Cortical Map Plasticity as a Function of Vagus Nerve Stimulation Intensity. *Brain Stimul.* **9**, 117–23 (2016).

Reviewers' comments:

Reviewer #1 (Remarks to the Author):

The paper reports interesting findings from a series of experiments that manipulations to enhance plasticity in central networks can improve motor and sensory recovery. They support the idea that CL-VNS could be an important new therapy to improve recovery after nerve damage.

There are some minor issues to be considered (below), but there is one major issue to be covered that could affect the results. Microstimulation methods are carried out in various ways and the results are influenced by the approach. Typically the effects are reported as the movement evoked at threshold of stimulation, which is usually in the range of 10s of μA given the parameters and electrodes used here. Elevating the threshold can change the form and type of movement and unmask other movements. Maps evident after one part of the body is disconnected could be altered if higher intensities are systematically used to reveal movements that would already be found in controls if these higher intensities were used. There is no description or discussion of the way the microstimulation was carried out in terms of intensities/thresholds used. Is the data always reported as threshold of stimulation? Where thresholds different in different groups (need to report in methods)? This is critical to address, as it could influence the results. The upper limit of stimulation reported (200) is very high- what were the typical/range actually used?. There needs to be detail in the methods about how this was done and in discussion, the effect of any differences in threshold needs to be considered for their impact on the results.

All of the below are minor comments that can be addressed without re-review and are meant to be constructive in helping make the report more effectively communicated:

Abstract could be much more informative about the nature of the question, methods used and the actual results, rather than the vague, general language sometimes used in this section. It would more more effectively communicate the impact and significance of the results, which could have a considerable clinical impact on the medical device community.

e.g. Rather than say 'severe damage', just say transection.

Line

48: the prognosis for full recovery of normal sensation and muscle control is poor (i.e., needs some qualification)

50: restore 'normal' function: most get partial recovery of function, some get abnormal sensations or control. This sentence implies that no recovery normally happens. Would be better if some qualification were added.

55 the word drama means "exciting, emotional, or unexpected series of events or set of circumstances." I think the authors mean 'large'?

71 reestablish normal (or original) central network. Perhaps adds clarity?

74: the motivation for CLVNS is obscure. The space of possible choices is very large. Could the authors motivate this choice here? Similarly, the choice of the cholinergic system would benefit from something that provides a rationale for choosing this system rather than waiting until the discussion. There are valid reasons for this selection.

85: without influencing peripheral nerve or muscle health: better as, in the absence of evidence for effects on ...nerve or...health." The statement now is beyond the data.

102; I think the evidence is stronger than 'suggests', I think 'indicates' fits the result appropriately.

104 paired with 'volitional' forelimb movements (to make it clear that the experimenters didn't do the movement, as might be done in therapies for humans).

107: could you add the number of pulses this is?

114-117. The maps are not clear enough to see if there are any points related to flexion

111 why was this expected? Do you mean as previously demonstrated?

114 p values should all be in similar format e.g., ($p < 0.0005$). Rather than reporting the p value which could be in methods as a threshold for acceptance (and perhaps in figs), it might be more interesting to report the % change in area across the results section. The level of confidence in the result (after you reach your accepted threshold) is less informative to the point of the study than the amount of change. This could also help to eliminate qualitative assessments as in line 124 ('substantial reduction').

129: is matched amount providing same amount of total current, number of pulses? Can this be explicitly stated. This will strengthen the paper in my view.

138-9. Multijoint movements can always be evoked from motor cortex if the intensity of stimulation is raised. Do the authors mean at the same intensity, with the same threshold?

156 rather than use the vague term 'substantially' why not give actual data on the % recovery. It is impressive and a number here would give the reader a measure of effect here in the text that one would have to extract from the figure.

This is a 'bug' I have (I am not alone), but I'd rather see the idiomatic expression "suffer from" removed and changed to something more evaluative, like "have chronic, debilitating problems...". This anachronistic expression conveys an emotional tone that, to me, is outside of what is appropriate for objective scientific communication. I would prefer to see it removed.

Reviewer #2 (Remarks to the Author):

Thank you for addressing all the questions.

Reviewer #3 (Remarks to the Author):

Very nice response to the reviewer comments.

Kevin Otto

Dear Editors and Reviewers,

We thank you for the opportunity to further revise our manuscript entitled “Enhancing plasticity in central networks improves motor and sensory recovery after nerve damage” (NCOMMS-19-07045). We have incorporated changes based on the reviewers’ comments, and we believe the manuscript is ready for publication. Below we address the reviewers’ concerns item by item. Blue text in the revised main manuscript indicates changes.

Reviewer #1

- 1. There are some minor issues to be considered (below), but there is one major issue to be covered that could affect the results. Microstimulation methods are carried out in various ways and the results are influenced by the approach. Typically the effects are reported as the movement evoked at threshold of stimulation, which is usually in the range of 10s of μA given the parameters and electrodes used here. Elevating the threshold can change the form and type of movement and unmask other movements. Maps evident after one part of the body is disconnected could be altered if higher intensities are systematically used to reveal movements that would already be found in controls if these higher intensities were used. There is no description or discussion of the way the microstimulation was carried out in terms of intensities/thresholds used. Is the data always reported as threshold of stimulation? Where thresholds different in different groups (need to report in methods)? This is critical to address, as it could influence the results. The upper limit of stimulation reported (200) is very high- what were the typical/range actually used?. There needs to be detail in the methods about how this was done and in discussion, the effect of any differences in threshold needs to be considered for their impact on the results?***

We thank the reviewer for raising these points on the ICMS mapping procedures, and we agree that the Methods section could benefit from additional description and rationale of the microstimulation mapping techniques. Here, we utilized long-duration intracortical microstimulation techniques to observe complex movements of the forelimb, as described in previous studies^{1,2}. Short-duration ICMS evokes small twitches in the forelimb, which can make accurate classification of paw movements, particularly differentiating between digit flexion and wrist extension, difficult. The long-duration procedures used in this study improved differentiation and categorization of these movements.

We observed stimulation thresholds comparable to previous studies utilizing long-duration ICMS^{1,2}. Notably, no differences were detected in the average movement thresholds across groups in the present study, all of which were near 100 μA and similar to uninjured animals (Supplementary Figure 2b). Furthermore, analysis of stimulation thresholds to evoke multi-joint movements reported in Figure 2c were not significantly different across groups (Supplementary Figure 4). Anecdotally, we observed that the movement elicited at threshold and 50% above

threshold were the same; however, higher stimulation intensity evoked movements that were more apparent and thus easier to accurately classify.

To address these concerns, we have we have added two supplementary figures depicting the average ICMS thresholds and the multi-joint ICMS thresholds (Supplementary Figure 2 & 4), and we now direct readers to these Supplementary Figures (Results, line 140). We have also revised the Methods section to provide more detailed descriptions of the intracortical microstimulation mapping procedures, and to provide additional rationale and justification for the long duration ICMS technique.

Abstract could be much more informative about the nature of the question, methods used and the actual results, rather than the vague, general language sometimes used in this section. It would more more effectively communicate the impact and significance of the results, which could have a considerable clinical impact on the medical device community.

e.g. Rather than say 'severe damage', just say transection.

Thank you for the feedback. We have modified the abstract as suggested.

Line

48: the prognosis for full recovery of normal sensation and muscle control is poor (i.e., needs some qualification)

We appreciate the feedback, and have modified the statement accordingly.

50: restore 'normal' function: most get partial recovery of function, some get abnormal sensations or control. This sentence implies that no recovery normally happens. Would be better if some qualification were added.

We have modified the statement accordingly.

55 the word drama means "exciting, emotional, or unexpected series of events or set of circumstances." I think the authors mean 'large'?

We appreciate the feedback and have replaced the word dramatic with extensive.

71 reestablish normal (or original) central network. Perhaps adds clarity?

We have modified the statement as suggested.

74: the motivation for CLVNS is obscure. The space of possible choices is very large. Could the authors motivate this choice here? Similarly, the choice of the cholinergic system would benefit from something that provides a rationale for choosing this system rather than waiting until the discussion. There are valid reasons for this selection.

We have modified and restructured the paragraph to clarify the motivation of CL-VNS.

85: without influencing peripheral nerve or muscle health: better as, in the absence of evidence for effects on ...nerve or...health.” The statement now is beyond the data.

We have modified the statement as suggested.

102; I think the evidence is stronger than ‘suggests’, I think ‘indicates’ fits the result appropriately.

We have modified the statement as suggested.

104 paired with ‘volitional’ forelimb movements (to make it clear that the experimenters didn’t do the movement, as might be done in therapies for humans.

We now include volitional to clarify the point raised by the reviewer.

107: could you add the number of pulses this is?

We now specify that there are 16 pulses per stimulation train.

114-117. The maps are not clear enough to see if there are any points related to flexion

We now include a reference to Supplementary Figure 2 so readers can compare bubble plots across groups including Uninjured subjects.

111 why was this expected? Do you mean as previously demonstrated?

We have clarified the statement.

114 p values should all be in similar format e.g., ($p < 0.0005$). Rather than reporting the p value which could be in methods as a threshold for acceptance (and perhaps in figs), it might be more interesting to report the % change in area across the results section. The level of confidence in the result (after you reach your accepted threshold) is less informative to the point of the study than the amount of change. This could also help to eliminate qualitative assessments as in line 124 (‘substantial reduction’).

We appreciate the feedback on reporting of effects between groups, and have amended the results section to present data as a percent change, where appropriate. To adhere to the journal format, we have elected to report exact p values.

129: is matched amount providing same amount of total current, number of pulses? Can

this be explicitly stated. This will strengthen the paper in my view.

We now specify that the stimulation is equivalent across groups.

138-9. Multijoint movements can always be evoked from motor cortex if the intensity of stimulation is raised. Do the authors mean at the same intensity, with the same threshold?

We observed a greater proportion of multi-joint movements in the Rehab and Delayed VNS groups compared to uninjured and CL-VNS groups at the same stimulation thresholds. We now clarify this in the text.

156 rather than use the vague term ‘substantially’ why not give actual data on the % recovery. It is impressive and a number here would give the reader a measure of effect here in the text that one would have to extract from the figure.

We have modified that statement and now report the percent recovery for each group.

This is a ‘bug’ I have (I am not alone), but I’d rather see the idiomatic expression “suffer from” removed and changed to something more evaluative, like “have chronic, debilitating problems...”. This anachronistic expression conveys an emotional tone that, to me, is outside of what is appropriate for objective scientific communication. I would prefer to see it removed.

We appreciate the feedback on this topic from the reviewer and have modified those statements.

References

1. Ramanathan, D., Conner, J. M. & H. Tuszynski, M. A form of motor cortical plasticity that correlates with recovery of function after brain injury. *Proc. Natl. Acad. Sci.* **103**, 11370–11375 (2006).
2. Ganzer, P. D. *et al.* Closed-loop neuromodulation restores network connectivity and motor control after spinal cord injury. *Elife* **7**, 1–19 (2018).